# The Molecular Biology of Placental Transport of Calcium to the Human Foetus

**DOI:** 10.3390/ijms26010383

**Published:** 2025-01-04

**Authors:** Valerie Walker

**Affiliations:** Department of Clinical Biochemistry, University Hospital Southampton NHS Foundation Trust, Southampton General Hospital, Southampton SO16 6YD, UK; valerie.walker@uhs.nhs.uk; Tel.: +44-2381206436

**Keywords:** Ca^2+^-calmodulin, DOHaD, NHERF1/EBP50, phospholipase C, plasma membrane Ca^2+^ ATPase, SOCE, TRPV6, pregnancy-specific beta-1-glycoproteins

## Abstract

From fertilisation to delivery, calcium must be transported into and within the foetoplacental unit for intracellular signalling. This requires very rapid, precisely located Ca^2+^ transfers. In addition, from around the eighth week of gestation, increasing amounts of calcium must be routed directly from maternal blood to the foetus for bone mineralisation through a flow-through system, which does not impact the intracellular Ca^2+^ concentration. These different processes are mediated by numerous membrane-sited Ca^2+^ channels, transporters, and exchangers. Understanding the mechanisms is essential to direct interventions to optimise foetal development and postnatal bone health and to protect the mother and foetus from pre-eclampsia. Ethical issues limit the availability of human foetal tissue for study. Our insight into the processes of placental Ca^2+^ handling is advancing rapidly, enabled by developing genetic, analytical, and computer technology. Because of their diverse sources, the reports of new findings are scattered. This review aims to pull the data together and to highlight areas of uncertainty. Areas needing clarification include trafficking, membrane expression, and recycling of channels and transporters in the placental microvilli; placental metabolism of vitamin D in gestational diabetes and pre-eclampsia; and the vascular effects of increased endothelial Orai expression by pregnancy-specific beta-1-glycoproteins PSG1 and PSG9.

## 1. Introduction

Three essential specifications for a system that delivers Ca^2+^ to the foetoplacental unit are as follows: First, it must be able to provide Ca^2+^ in a stringently titred amount for Ca^2+^ signalling in the placenta and foetus, starting at fertilisation and continuing to term. Second, it must be able to simultaneously provide a flow-through system to transfer increasing amounts of Ca^2+^ safely across the placenta to mineralise foetal bones without increasing the cytosolic Ca^2+^ to toxic levels. By approximately 35 weeks of gestation, around 300 mg of Ca^2+^ is transferred daily [1]. Third, it must match the supply to the changing and increasing needs of the foetus [2,3,4]. The source of Ca^2^ is the mother. Before around 16 weeks of gestation, Ca^2+^ is provided by uterine fluid, which contains secretions from decidual glands in the uterine endometrium [5,6]. The Ca^2+^ content of this fluid in humans is unknown; however, in sheep, it increased from 14 days postfertilisation (pf.) [7]. As the mature placenta forms and begins to function, the source is maternal blood bathing the foetal placental villi. With an average ionised Ca^2+^ concentration of approximately1.18 mM [8], this is around 11,800 times higher than the normal resting intracellular cytosolic concentration (<100 nmol/L) [9]. To meet these demands, the surface of the foetal placenta is covered by a very large syncytium of trophoblasts, covering around 5 m^2^ at 28 weeks of gestation and increasing to 11–12 m^2^ at term, which is folded into a myriad of branched free-floating villi [10,11]. Embedded in the villus membranes are a body of Ca^2+^ channels, which collectively can respond to a full range of stimuli from growth factors, cytokines, hormones, and mechanical stress [12,13]. Failure to meet the above requirements may result in ectopic pregnancy [14], growth restriction with risks to the foetus in utero and postnatally, early pre-eclampsia (PET) with risks to mother and foetus [3,15,16], and possibly long-term risks for osteoporosis in adult life [17,18].

We need to understand the mechanisms involved to intervene appropriately to reduce the risks from malfunction and to protect the bones of infants born extremely preterm and/or with birth weights < 1000 g [19,20]. Because of ethical issues, data on primary human tissues are scarce. Investigation requires a multi-pronged approach, drawing in data from human genetic disorders; an increasing range of animal models; physiological studies of trophoblast cell lines; tumour tissues; basic science; and, increasingly, gene expression studies using RNA sequencing (RNA-Seq) and chromatin immunoprecipitation-sequencing ChIP-Seq studies to investigate gene regulation. Extrapolation of findings from animal models may be problematic because of differences in placental structure and gene expression, notably between two commonly used animals, namely, sheep for physiological experiments [21,22,23] and mice for genetic manipulation [24,25]. Further, experimental procedures to collect samples for analysis influence the results [1]. RNA-Seq and microarrays only show genes expressed at a single time point and will differ according to epigenetic regulation. This accounts for some of the wide intra- and inter-individual variation and smaller racial differences observed [26,27]. Reported cell studies have used cell lines from choriocarcinomas, except for HTRB/SV neo, a human embryonic villus cell line transfected with simian virus 40 antigen. However, a new human trophoblast line has been produced [28], and others must follow. The use of 3D organoids that resemble the structure and physiology of the human placenta will enable studies of developmental events in human implantation and placentation [29,30,31,32,33].

Our insight into the mechanisms of placental Ca^2+^ transport has extended rapidly because of recent advances in analytical, genetic, and computer technology [34,35,36,37]. Coming from a variety of different sources, the published data are scattered. The aims here are to look for common threads and to highlight deficiencies in our knowledge. This may help to guide future research and policies for interventions to improve bone health. The items covered were selected because of their known involvement in Ca^2+^ turnover postnatally or because they have been shown experimentally to affect placental Ca^2+^ transport. There are four sections: Section A describes placental development, and Section B covers calcium carriers and transporters that control calcium. The transient receptor potential vanilloid 6 (TRPV6) channel is covered in depth. This has emerged as the major Ca^2+^ import channel in the placental villi, and its importance is manifest in human inherited TRPV6 deficiency. Section C covers the effects of peptides and hormones on placental calcium transport, and Section D is a brief overview of the developmental origins of health and disease (DOHaD) proposal in relation to postnatal bone development.


**Section A: The Placenta: Laying the Foundation**


## 2. Placental Development

A primitive placenta forms by approximately 10–12 weeks pf., but the villi have a low surface area and are poorly vascularised [38]. The definitive placenta has developed and functions by approximately 20 weeks pf. and grows exponentially until term [5,6]. The very early processes through which the embryo transfers from the uterine fallopian tube and becomes engulfed (implanted) within the endometrium of the body of the uterus are of paramount importance for normal development of the placenta. Figure 1 shows some key intermediates [6,39,40,41]. These are summarised briefly in Table 1, which also indicates the likely sources of calcium at each stage. For detailed information, refer to the literature [5,6,41,42,43,44].

At term, the placenta is 15–20 cm in diameter, weighs around 500 g, and normally has a centrally placed umbilical cord [45]. Septa arising from the basal layer of the uterine decidua grow into the foetal placenta, dividing it into 15–20 cotyledons, which each contain an anchoring (stem) villus and the villi that branch out from it—the intermediate and terminal villi. The highly branched terminal villi are the placental workforce (Figure 2).

Spiral branches of the maternal uterine arteries empty into the intervillous spaces. Blood percolates around the villi and then leaves via uterine veins. Two foetal arteries carry blood from the villi to the foetus via the umbilical cord, where they lie alongside a single vein carrying blood from the foetus to the maternal venous system. There is no continuity between maternal and foetal circulations in an undamaged placenta [40,41].

### Terminal Villi: Actin, Ezrin, and NHERFI/EBP50

At term, the terminal villi are mobile, branched, and dissimilar from the rigid, unbranched microvilli of the duodenal mucosa. They have a core of bundled actin filaments, which are cross-linked and extend only a short distance into the cell cytoplasm. They are attached to the plasma membrane directly or indirectly, in part by ezrin [46,47,48,49]. At low cytoplasmic Ca^2^ concentrations, a villus protein, namely, α-actinin, probably crosslinks the actin fibres, reinforcing the cytoskeleton. At Ca^2+^ concentrations > 0.3 μM, the release of α-actinin would weaken the structure [48]. Two abundant villus proteins, namely, ezrin and Na^+^/H^+^ exchange regulatory cofactor 1 (NHERF1), interact and bind with actin. Figure 3 is a hypothetical model of the terminal villus core based on the better-defined structure of duodenal villi [50].

Ezrin, belonging to the ezrin, radixin, and myosin (ERM) protein family, accounted for approximately 5% of the total mass of protein isolated from placental syncytiotrophoblast and was present mainly as noncovalent dimers and higher-order oligomers [51]. ERMs regulate microvillus formation in tissue culture epithelial cells. Cells lacking ERMs have reduced numbers of microvilli [52]. Ezrin contains an NH_2_-terminal (N) domain of around 300 residues and a 100-residue COOH-terminal (C) domain. In a dormant form in the cytoplasm, the C-terminal tail binds to the N-terminal FERM domain and closes the molecule [53,54]. Phosphorylation of critical residues at the interface between the N- and C-terminal regions blocks this association and stabilises ezrin in an open state. The freed C-terminal binds to actin [55], and the N-terminal is directed towards ERM binding proteins associated with the apical membranes of the microvilli.

A search for ERM binding partners in placental microvilli identified a protein named as ezrin binding protein 50 kDa (EBP50) [56]. This colocalised with actin and ezrin and specifically associated with the microvilli of the placental syncytiotrophoblast. EBP50 was also found in cultured JEG-3 human choriocarcinoma cells [51,57]. Because EBP50 has two postsynaptic density 95/discs large/zonula occludens-1 (PDZ)-binding domains, which associate with integral membrane proteins, it was proposed that EBP50 might mediate the membrane attachment of ezrin. The microvilli of EBP50-null mice are short and abnormal, like those of ezrin-null mice, suggesting that the two proteins function together in microvillus structure or regulation [58,59]. However, this possibility does not appear to have been investigated. Independently, others identified the same protein through its involvement with regulation of the rabbit renal brush border Na^+^/H^+^ ion exchanger, NHE3 [60,61]. This protein was named NHERF1, and EBP50 has been renamed. NHERF1 has two NH_2_-terminal PDZ domains, PDZ1 and PDZ2, and a carboxyterminal ERM binding domain (Figure 4) [62,63]. The ERM binding site and a cholesterol-binding site in PDZ1 promote its close association with the villus membrane.

NHERF1 (solute carrier family 9 member A3 regulator 1, SLC9A3 regulator 1, gene *NHERF1*) is a cytoplasmic multifunctional scaffolding protein. It scaffolds membrane-bound proteins to the sub-apical actin cytoskeleton and stabilises them at the cell surface [62,63,64]. However, its main role is to connect membrane-associated proteins with transiently assembled cytosolic complexes, including kinases, phosphatases, and trafficking proteins, to direct cell signalling or transport activities [56,62]. Four members of the NHERF family are expressed in the kidneys (NHERF1-4). Whether the placenta expresses all forms is unknown. Amongst numerous membrane proteins bound by NHERF1 are the sodium–phosphate cotransporters NaPi-2a and NaPi-2c, and NHE3, which are expressed in placental microvilli. NHERF1 is phosphorylated by several kinases, which alter its binding activity and downstream signalling events [65,66]. Its role in the placenta is unknown.


**Section B: Tools for Controlling intracellular Ca^2+^: Ca^2+^ Channels and Transporters**


## 3. Sources and Removal of Cytosolic Ca^2+^

Ca^2+^ enters the cytosol through two routes: (i) from the extracellular space via channels or transporters in the plasma membrane, (Section 4) or (ii) by release from stores in the endoplasmic reticulum (ER). Rapid increases in Ca^2+^ trigger signalling, which is then cascaded through the cell along specified pathways to generate an appropriate response [67]. Surplus Ca^2+^ must be cleared rapidly to avoid a global increase in cytosolic Ca^2+^ and widespread uncontrolled stimulation of signalling (Section 6). A collection of Ca^2+^ entry channels is expressed variably on placental membranes across gestation. These include members of the transient receptor potential (TRP) family [12,68]. Current evidence indicates that the TRPV6 channel is the major channel mediating the vast influx of Ca^2+^ through terminal microvilli in the third trimester (Section 4.2). Surges of incoming Ca^2+^ through plasma membrane channels can increase the Ca^2+^ concentration across the cell to 0.5 to 1 μM, but concentrations may exceed 100 μM at the channel mouth [67]. It is essential that this is directed directly to the relevant signalling complex assembled close to the channel and that channel opening and closure are tightly regulated to respond promptly to changes in stimulation. Intracellular Ca^2+^ release from ER stores is mediated by inositol trisphosphate (IP3), generally generated by phospholipase (PLC) activation in response to stimulation of receptors in the plasma membrane (Section 3.1).

### 3.1. Phospholipase C

The ER acts as an intracellular store of Ca^2+^, which is readily available for a rapid signalling response to stimulation of membrane receptors. ER Ca^2+^ concentration in resting cells is normally around 400 μM [69], ranging from 200 μM to 650 μM [9]. Ligand binding to G-protein-coupled receptors, tyrosine kinase receptors, or other plasma membrane receptors activates phospholipase C (PLC) [70]. There are six PLC families, β, γ, δ, ε, η, and ζ, which all have four Ca^2+^-binding EF hands and catalytic domains. They are all targeted to membrane-bound phosphatidyl inositol 4,5 bisphosphate (PI (4,5) P2, alias PIP2). Except for PLCζ, this is via an amino-terminal pleckstrin homology (PH) domain [70,71,72,73]. PLC cleaves PIP2, releasing IP_3_ (Figure 5).

IP_3_ activates the inositol trisphosphate receptor (IP_3_R) on the surface of the ER, leading to release of Ca^2+^ into the cytosol. There are three IP_3_R isoforms, denoted IP_3_R1–3, that function as tetramers. Each monomer has a cytosolic amino-terminal domain that binds IP_3_; a regulatory domain that binds Ca^2+^, ATP, and other modulatory molecules/proteins; and a carboxy-terminal channel that contains six transmembrane domains and a short cytosolic tail. The activation and opening of the IP_3_R require binding by both Ca^2+^ and IP_3_. Dual regulation enables the channel to support long-lasting Ca^2+^ oscillations [73].

### 3.2. PLCζ (Zeta)

PLCζ is only expressed in sperm heads in mammals. In humans, PLCζ localises to three distinct regions [74,75,76,77]. Following the fusion of sperm with the egg, PLCζ releases IP_3_ from PIP2. This mediates Ca^2+^ release from the ER, causing oscillations in cytosolic Ca^2+^ that activate the ovum via numerous pathways, concluding in cortical granule exocytosis, resumption of meiosis II, and pronuclear formation. Cortical granule exocytosis, mediated principally by protein kinase C (PKC), releases enzymes that modify the zona pellucida and prevent further sperm entry. PLCζ is the only protein known so far that initiates Ca^2+^ oscillations during human fertilisation [73]. Ca^2+^-bound Ca^2+^-calmodulin-stimulated protein kinase II (CAMKII) ensures proper cell cycle progression. In the absence of Ca^2+^ at fertilisation, embryos consistently display a reduced inner cell mass at the blastocyst stage [78]. Oocyte activation deficiency (OAD) is the basis of total fertilisation failure (TFF) and is attributed to mutations in the PLCζ gene, termed male factor infertility [70,79].

## 4. Ca^2+^ Importation Across the Plasma Membrane

### 4.1. Transient Receptor Potential Channels (TRPs)

TRPs are widely distributed ion channels, which are permeable to monovalent and divalent cations, including Ca^2+^, Mg^2+^, Na^+^, and K^+^, with highly variable Ca^2+^/Na^+^ permeability ratios [67,80,81,82,83]. They act as sensors of chemically toxic and physical stimuli [84]. Mammals express 28 TRP channels that compose the TRP superfamily. There are six subfamilies based on amino acid sequence homology [85], designated as canonical (TRPC), vanilloid (TRPV), melastatin (TRPM), ankyrin (TRPA), mucolipin (TRPML), and polycystin (TRPP). TRPCs fall into four subsets: TRPC1, TRPC2, TRPC3/6/7, and TRPC4/5 [83,86]. TRPC2 is present in rodents but is a pseudogene in humans [87]. TRPV channels were named for their sensitivity to vanilloid and capsaicin [88,89]. TRPV5 and TRPV6 differ from all other TRP channels because of their high Ca^2+^ selectivity (permeability of Ca^2+^/Na^+^ > 100) [90,91,92,93]. TRPM6 and TRPM7 are permeable to both Ca^2+^ and Mg^2+^, but their main role may be to regulate Mg^2+^ import because they are sensitive to physiological Mg^2+^-ATP concentrations. Unlike the other TRPs, they have a kinase domain [94,95]. Although *TRPV5/6* are reported to be co-expressed in the human placenta [96,97,98], expression of *TRPV5* is negligible and 1000-fold lower than that of *TRPV6* [82,99]. *TRPV6* is expressed in the human placenta at term [16,100], in human trophoblasts and syncytiotrophoblasts [101], and in the mouse placenta [102].

TRPs assemble as homotetramers or frequently heterotetramers to form a cation-conducting pore [12,80,83,103,104]. Each monomer has six transmembrane domains (S1–S6) and intracellular NH_2_ and COOH (C) termini of variable length. Members of the TRPC, TRPM, and TRPV subfamilies have a conserved TRP domain comprising approximately 25 amino acids in the C-terminal. The function of the TRP domain is uncertain, but it may be involved in PIP2 binding [105] or subunit assembly [106]. Multiple regulatory elements, interaction sites, and enzymatic domains are present in the intracellular termini [80]. Diacylglycerol (DAG) is a direct activator of TRPC3, TRPC6, and TRPC7 channels [83,107]. Ankyrin repeat domains are present on the NH_2_ terminus of TRPC, TRPV, and TRPA subunits. The COOH terminus of TRPC channels has calmodulin (CaM)/IP_3_R-binding (CIRB) domains; Ca^2+^-binding EF hands; and in the case of TRPC4, a PDZ domain [81]. TRPs regulate many signalling pathways, including the mitogen-activated protein kinase (MAPK), transforming growth factor (TGF)-β, nuclear factor kappa-B (NF-κB), and AMP-activated protein kinase (AMPK) pathways. TRP channel activation induces multiple immunomodulatory effects [108,109].

### 4.2. Transient Receptor Potential Vanilloid 6 (TRPV6)

TRPV6 (alias CAT1) was identified in rat duodenal cells and shown to mediate intestinal Ca^2+^ absorption [110,111]. A related channel, TRPV5 (alias CATII), is expressed predominantly in the kidney and mediates trans-cellular Ca^2+^ reabsorption by a well-defined pathway [112]. Its primary function is to regulate urinary Ca^2+^ excretion [113,114]. They share 75% sequence homology and, unique for TRPs, have very high selectivity for Ca^2+^ ions. Unlike TRPV5, TRPV6 is expressed widely in normal epithelial tissues. Vitamin D, oestrogen, and dietary Ca^2+^ regulate the abundance of TRPV6 in the duodenum [115]. There is mounting evidence that *TRPV6* expression is increased, or suppressed, by epigenetic mechanisms [116,117]. Two initiation sites for *TRPV6* transcription have been identified: (i) a classic AUG methionine start codon and (ii) an unusual, non-canonical ACG codon, located 120 base pairs upstream of the AUG start site [118], which is decoded as methionine and not threonine as predicted. This produces a protein transcript with a 40-amino-acid extension on the N terminus, which may be important for attachment to the plasma membrane. *TRPV6* expression in the placenta is driven largely by oestrogens [111,115]. TRPV6 is expressed by syncytiotrophoblasts from human term placenta [111,118,119,120,121,122]. Unlike the duodenum [116], 1,25(OH)_2_D_3_ did not increase *TRPV6* expression significantly [100]. In mice, *trpv6* was expressed at E10 pf. and increased by 14-fold during the last 4 days of gestation [123]. In rats, expression increased steadily and peaked at E20.5 pf. Expression was found to be progesterone- and oestrogen-receptor dependent [124] and increased in hypoxia [125]. *Trpv6* mRNA and protein were also expressed in mouse bone, with a gradual increase from E9.5 pf. to E15.5–17.5 pf. but a marked decrease at parturition [123]. Unlike human and rodent placentas, in ungulates the cotyledons (referred to as placentomes) are not aggregated but are implanted separately in the endometrium and are connected by foetal membranes [126]. *Trpv6* mRNA was found to increase gradually in the placentomes of cows through gestation but without an observable increase in proteins. In contrast, there was a large increase in both mRNA and protein of TRPV6 in the membranes, evidence that maternofoetal Ca^2+^ transport mainly occurs in intraplacentomal regions [126]. It seems that the processes of Ca^2+^ transfer for signalling in the placenta and provision for foetal bones are largely segregated in cows. Kogel, Fecher-Trost, Wissenbach, et al. (2022) [127] observed that in living cells, TRPV6 had a very short residence time in the plasma membrane and was barely discernible. This was explained by clathrin-mediated internalisation triggered by locally released Ca^2+^. TRPV6 was demonstrated in Rab7-positive endosomes targeted for recycling to the membrane or (most) in Rab11a-positive late endosomes for lysosomal degradation.

#### 4.2.1. Structure and Operation

TRPV6 channels are constitutively open, allowing electro-chemical gradient-driven Ca^2+^ entry into the cell [128,129], but they are inactivated reversibly by Ca^2+^ to protect from a toxic influx. The *K*_m_ value for Ca^2+^ uptake was 0.25 mmol/L for human TRPV6 [130]. Mg^2+^, gadolinium Gd^3+^, ruthenium red, low extracellular pH, 2-aminoethoxydiphenyl borate (2-APB), and soricidin derivatives decrease flow through the channel [82,115,131]. Like the other TRPs, the TRPV6 channel is tetrameric, with the pore located centrally and formed by helix 6 of all four monomers, as described in detail in the literature [132,133,134]. Figure 6 illustrates the important domains. Features of relevance to channel closure are the intracellular “skirt” formed by the interlocking N and C termini, which encloses a 50 Å × 50 Å wide cavity below the pore of the ion channel; the molecular cage at the pore orifice formed by four TRPV6 W583 residues, one from each monomer; and binding sites for Ca^2+^-CaM in the C-terminal tails.

Closure of the channel by Ca^2+^ inactivation has two components. An initial fast inactivation is attributed to amino acid sequences in the intracellular loop between trans-membrane helices H2 and H3 [135] and in the helix–loop–helix (HLH) domain of the TRPV6 N-terminal [136]. A slower process involves Ca^2+^/CaM binding to the TRPV6 C-terminal. CaM is a monomeric protein comprising two structurally similar globular lobes at the N- and C-terminals connected by a linker helix. Each lobe has two Ca^2+^-binding EF hands. The C-lobe binds Ca^2+^ more tightly (Kd around 0.2 μM) than the N-lobe (Kd around 2 μM) [137,138,139]. The N-lobe only becomes Ca^2+^ loaded when the intracellular Ca^2+^ increases significantly above the resting concentrations (upper limit approximately100 nM [140]), and hence, it can act as a Ca^2+^ sensor. In the absence of Ca^2+^, apo-CaM adopts a closed conformation. When fully loaded, the inter-lobe linker is extended so that the molecule opens and becomes flexible, and the N- and C-lobes are distanced [140,141,142]. The lobes bind at separate sites on the TRPV6 carboxy terminal. This causes the CaM C-lobe to rotate towards the channel pore, and CaM K115 engages with the W583 molecular cage at the pore orifice. This interaction not only blocks the channel exit but also initiates a sequence of intramolecular events that prevent Ca^2+^ entry into the TRPV6 channel [115,140,142,143,144,145].

#### 4.2.2. Associated Proteins

In addition to CaM, other proteins that bind or associate with TRPV6 are S100 calcium-binding protein A10 (S100A10) [142,146], Na^+^/H^+^ exchange regulatory cofactor 4 (NHERF4) [147], Ras-related protein Rab-11A (Rab11a) [148], TRPM8 channel-associated factor 2 TCAF2 [149], cyclophylin B [150,151], klotho [98], and TRPC1 [152,153]. S100A10 increases expression of TRPV6 at the cell surface (Section 4.2.4). Rab 11a is required for endosomal membrane fusion and may have a role in recycling TRPV6 at the cell surface [154]. Cyclophilin B associates with TRPV6 in human syncytiotrophoblasts and increases TRPV6 activity in vitro [82]. It regulates the activation of interferon-regulatory factor 3 [IRF3]. Klotho has low expression in the placenta. TRPC1 and TRPV6 interact via their N-terminal ankyrin repeats, resulting in downregulation of TRPV6 expression in the plasma membrane and decreased TRPV6 Ca^2+^ currents, perhaps by retaining TRPV6 in intracellular compartments [153].

#### 4.2.3. TRPV6 Deficiency in Pregnancy

Trpv6 deficiency occurs in human foetuses with inherited mutations of the *TRPV6* genes and has also been associated with pre-eclampsia (foetal and placental abnormalities were observed in mouse models with inactivating mutations of the *trpv6* gene).

Inherited TRPV6 deficiency, namely, transient hyperparathyroidism of the newborn, OMIM 618188, has been reported in 12 individuals [115,155,156,157,158,159]. Inheritance was autosomal recessive, and there was similar sex distribution. Most were born at term, and five had low birth weight. Eleven had a gross disturbance of bone turnover (bone dysplasia), which was evident in utero in the third trimester. At birth, this manifests as a narrow, funnel-/bell-shaped chest; soft abnormal ribs with fractures; short, sometimes bowed, femora; and a generalised deficiency of mineralised bone (osteopenia). There were no other physical abnormalities. The phenotype was normal in the twelfth case. The chest restriction caused severe life-threatening respiratory problems and feeding difficulties requiring intensive support, often for several months. At birth, five babies had a low serum-ionised or corrected calcium. All had raised parathyroid hormone (PTH) levels. The serum 25(OH)D of seven newborns indicated vitamin D insufficiency. Remarkably, after the correction of serum Ca^2+^, the skeletal abnormalities resolved over months, with normal oral intakes of calcium and vitamin D supplements for infancy [155,156,157,158,159,160]. Most mutations were inactivating mutations in ankyrin repeats, transmembrane helices or their linkers, or the C-terminal hook but not in the channel pore. However, two were gain of function mutations located at fast Ca^2+^ inactivation sites in TRPV6 [135,136].

Additional observations relevant to the pathogenesis of this defect were as follows: (i) A bone abnormality was detected at 20 weeks of gestation in one foetus, which progressed during the third trimester [155]. This early manifestation might be explained by local effects of bone TRPV6 deficiency, rather than poor placental Ca^2+^ provision. (ii) Four of the five affected newborns were Japanese. One contributory factor might have been a low maternal dietary Ca^2+^ intake [161,162]. The average Ca^2+^ intake in Japan has been estimated at approximately 480 mg/day, which is below the recommended intake for Asia of 700 mg/day [162]. The individual with a normal phenotype was Japanese and had the same mutations as an older severely affected sibling. It was suggested that the calcium status of the mother might have differed between pregnancies [158]. (iii) In one affected newborn, serum 25(OH)D indicated frank vitamin D deficiency, as did the concentrations of her mother and dizygous twin sibling. The twin had normal TRPV6 alleles. His serum Ca^2+^ was low, but he had no phenotypic abnormalities [157]. Hence, frank vitamin D deficiency alone did not cause overt bone abnormalities in utero. (iv) The placenta collected at term delivery of one of the affected foetuses [155] had a normal weight and macroscopic appearance [160]. However, compared with a control placenta from an unaffected pregnancy, 15 proteins were significantly increased, and 4 were decreased. Two proteases, HTRA1 (high-temperature requirement A serine peptidase 1) and cathepsin G, were only found in the affected placenta [163].

Pre-eclampsia is associated with foetal growth restriction [16]. Ca^2+^ transport by cultured primary syncytiotrophoblasts from placentas of women with pre-eclampsia was significantly lower than that of matched controls, as were mRNA and protein expression of genes encoding TRPV5, TRPV6, calbindin D9k, calbindin D28K, and plasma membrane Ca^2+^ ATPase-1 and -4 (PMCA1 and PMCA4). mRNA expression of *Ip3R* and the ryanodine receptor (*Ryr)* was also decreased, but mRNA of sarcoendoplasmic reticulum ATPase (*SERCA 1, 2* and *3*); the mitochondrial voltage-dependent anion channel 1 *(VDAC*); and the DNA repair gene 8-oxoguanine glycosylase (*OGG1*) were increased. The findings indicated a disturbance in placental Ca^2+^ homeostasis and transport, possibly attributable to ATP depletion and oxidative stress [16].

In mice, TRPV6 was inactivated by knock-down of the gene or by introducing a mutation at the TRPV6 channel pore [102,164]. TRPV6-deficient foetuses accumulated less Ca^2+^ [164]. Placentas from pregnancies in which both the mother and the foetus had homozygous mutations were structurally abnormal. The foetal labyrinth was thicker and less dense and had much larger cell-free spaces than in placentas from wild-type (WT) pregnancies [102]. Whereas WT-cultured placental trophoblasts connected with many adjacent cells and built a tight cellular network, placental cells from mutant animals had less contact with neighbouring cells, and their extracellular matrix (ECM) was less dense. Mutant trophoblasts had higher expression of some proteases, including HTRA1 and granzymes (cytotoxic serine proteases), and decreased fibronectin, a component of the ECM [164]. Zebrafish with a naturally occurring loss-of-function *trpv6* mutation [R304Stop] had a 68% reduction in total body calcium content, bone mineral defects, and a marked reduction in Ca^2+^ uptake by the yolk sac and gills [165]. Zebrafish TRPV6 protein is expressed in yolk sac cells, and long stretches of the amino acid sequence are closely homologous with the human TRPV6 sequence [166].

#### 4.2.4. TRPV6 Membrane Expression: S100A10 and Annexin A2

S100A10 (p11, annexin II light chain, calpactin light chain) is a multifunctional protein with a wide range of physiological activity. Unlike other S100 family members, it cannot bind Ca^2+^ but is locked in the Ca^2+^-loaded conformation. S100A10 interacts with several proteins but predominantly exists in a heterotetrameric complex with Annexin A2 (ANXA2) [114,167,168]. ANXA2, S100A10, and the heterotetramer were expressed in human syncytiotrophoblast, mainly in areas with active placentation [169]. Both ANXA2 and S100A10 were demonstrated in brush-border membrane vesicles from human placenta, and both increased progressively during gestation [170]. *ANXA2* and *S100A10* were among 115 genes in sequenced transcriptomes from the placentas of 14 mammalian species and were expressed in all and classed as core genes [25]. This suggests that they may have important roles.

Annexins are Ca^2+^-binding proteins that interact with biological membranes. The ANXA2 monomer is an intracellular protein with a core Ca^2+^- and phospholipid-binding domain and a small N-terminal tail domain [168,169]. The Ca^2+^-binding sites are distinct from EF-hand sites [171]. Ca^2+^ liganded by carboxyl and carbonyl oxygens located on the ANXA2 surface binds with negatively charged phospholipids, including PIP2 and phosphatidylserine with high affinity [172], and bridges ANXA2 to the cell membrane. In the presence of Ca^2+^, ANXA2 also binds to F-actin [168]. The first 14 N-terminal ANX2 residues bind with S100A10, forming a tight dimer, and two dimers combine to form a heterotetramer that interacts with actin and membrane phospholipids [114,168]. The S100A10 subunits face the cytosol and can interact with additional proteins, including ion channels such as TRPV6 [168]. The membrane binding function of ANXA2 in the complex may actuate endocytosis or exocytosis and/or the regulation of intracellular transport of various ion channels and transporters [167,168,169,173,174]. S100A10 associates with a conserved sequence in the C-terminal tails of TRPV5 and TRPV6 in a Ca^2+^-independent manner. The complex plays a crucial role in routing TRPV5 and TRPV6 to the plasma membrane. In the kidney, a significant subset of TRPV5 channels is localised subapically in the distal renal tubules, suggesting that TRPV channels may be shuttled to the plasma membrane [146,175]. Downregulation of ANXA2 inhibited TRPV5- and TRPV6-mediated currents in transfected HEK293 cells [114].

## 5. Store-Operated Ca^2+^ Entry (SOCE)

Recurrent stimulation of membrane receptors that activates PLC and generates IP_3_ drains Ca^2+^ from the ER stores (Section 3). This must be replenished from extracellular sources. A major route demonstrated in non-excitable tissues was via plasma membrane Ca^2+^ channels, which were activated by the depleted stores: store-operated Ca^2+^ entry (SOCE) channels, also termed capacitive Ca^2+^ entry channels [176,177]. These were characterised electrophysiologically and named the Ca^2+^ release-activated Ca^2+^ (CRAC) channel, but the protein responsible was unknown. They had a high selectivity for Ca^2+^ versus Na^+^, displayed an inwardly rectifying current–voltage relationship, and responded to a decrease in ER Ca^2+^ [69]. Clarson, Roberts, Hamark, et al. (2003) [178] demonstrated for the first time that SOCE occurred in term human placenta. Ca^2+^ depletion of term villi by incubation in a Ca^2+^-free buffer with thapsigargin to inhibit Ca uptake into the ER, followed by superfusion with Ca^2+^-containing buffer, resulted in a rapid rise in [Ca^2+^]_i_. This was inhibited by GdCl_3_, NiCl_2_, CoCl_2_, or the channel inhibitor MSKF96365 but not by nifedipine. This was not demonstrable in fragments of first-trimester placenta. mRNA encoding TRPC1, TRPC3, TRPC4, TRPC5, and TRPC6 was identified in both first-trimester and term placentas.

The human CRAC ortholog was identified in two independent studies using unbiased genome-wide RNA interference screens in Drosophila cells [179,180]. This was characterised as a plasma membrane–resident protein encoded by gene *FLJ14466* [179]. Feske, Müller, Graf, et al. (1996) [181] found that T lymphocytes from two siblings with one form of hereditary severe combined immune deficiency (SCID) syndrome were defective in store-operated Ca^2+^ entry and CRAC channel function. This rare disorder can be caused by mutations in a variety of genes that lead to impaired function of T, B, or natural killer cells. Using a combination of a modified linkage analysis with single-nucleotide polymorphism arrays and a Drosophila RNA interference screen, they subsequently identified a point mutation in the CRAC gene, which they named Orai1 [182]. Later, mutations of stromal interaction molecule 1 *(STIM1*) (Section 5.2) were identified in two siblings with SCID with defective store-operated Ca^2+^ entry [183]. The CRAC channels mediate SOCE through the partnership of Orai and STIM1 [184,185,186]. Several non-selective Ca^2+^-permeable TRPC channels (Section 5) have also been proposed to act as store-operated channels [185,186,187,188], although this is controversial [69]. TRPC1 was thought to be incorporated into a combined “ISOC” channel with Orai1 and STIM1. Although both TRPC1 and Orai1 interact with STIM1, there are no data to show direct interaction of TRPC1 with Orai1. The available evidence indicates that TRPC1 and Orai1 are distinct channels [185,186].

### 5.1. CRAC/Orai1 Entry Channel

Human Orai1 (CRAC) channels are around 55 Å long and narrow in the closed conformation [184]. They assemble and function as hexamers. Each subunit comprises four transmembrane helices, (TM1, TM2, TM3, and TM4). The hexameric channel contains a single ion conduction pore lined by the TM1 helices. The TM4 domains form an outer coat that interacts with the plasma membrane and STIM [189]. Incoming Ca^2+^ ions are concentrated by a ring of acidic E106 side chains. Below this, two rings of residues, V102 and F99, form a hydrophobic gate. Activation of the channel by STIM1 causes the pore helices to rotate, moving the F99 residues away from the pore axis and enabling Ca^2+^ ions to flow through [189].

The Orai1 channel is highly Ca^2+^-selective with a Ca^2+^:Na^+^ permeability ratio of >1000, but in the closed state, the unitary Ca^2+^ conductance is extremely low, estimated at around 10–35 fS [190]. Activation of the channels is not voltage-dependent, and they are insensitive to most of the common voltage Ca^2+^ channel inhibitors. Extracellular Ca^2+^ potentiates channel activity by several fold [69], and associated Orai-binding proteins including CaM may fine tune activity by unknown mechanisms [69]. CRAC activity is modulated directly by lipids and posttranslational modifications or indirectly by accessory proteins at the ER–plasma membrane contact site. The channels undergo fast Ca^2+^-dependent inactivation (milliseconds) due to binding of Ca^2+^ close to the intracellular face of the pore [69]. Over a longer timescale (seconds to tens of seconds), intracellular Ca^2+^ accumulation causes slow inactivation [69]. Refer to the literature [191,192,193,194] for more detail.

### 5.2. Stromal Interaction Molecule 1 and 2 (STIM1 and STIM2)

STIM1 and STIM2 are single-pass ER membrane proteins with >74% sequence similarity (54% sequence identity). Their functions are to sense the ER Ca^2+^ concentration and to respond to Ca^2+^ depletion by activating Orai1 and opening the channel to allow a large influx of Ca^2+^ into the cell. STIM1 and STIM2 work in concert and can form heterooligomers. STIM2 has a lower affinity for Ca^2+^ than STIM1 and is therefore more sensitive to small changes in ER Ca^2+^ concentration and can initiate an earlier response (reviewed [69,186,191,195]). STIM1 and STIM2 have a luminal NH_2_ terminus and a cytoplasmic COOH terminus, are glycosylated, and have a series of structural modules with defined functions [69,186,191,196,197]. Figure 7 shows the main domains for STIM1 and their roles.

In the N-terminal within the ER, there are two Ca^2+^-binding EF-hand domains, EF1 and EF2, and a sterile alpha motif (SAM), which is essential for the dimerization/multimerization of the STIM monomers and activation of the protein [191]. The STIM-Orai activating region (named SOAR) is located at two of the three highly conserved coiled-coil (CC) domains, CC2/CC3, on the cytosolic side of the ER membrane [69,189,198]. Positively charged lysine residues in the polybasic domain (PBD) at the C-terminal interact electrostatically with acidic phospholipids and acyl chains in the plasma membrane, including PIP2, and localise the activated STIM proteins at the ER–plasma membrane (PM) [186].

In the resting state, the two monomeric EF-hand domains bind five or six free Ca^2+^ atoms within the ER [189]. The STIM monomers are not associated, and their N-terminals float freely. The CC1 coil binds with the CC2/CC3 SOAR domain, blocking the access of the STIM to Orai1 and the attachment of the PBD to the plasma membrane. When ER Ca^2+^ is depleted and the EF-binding sites are unoccupied, there is a dramatic change in the STIM conformation, initiated by the dimerization/polymerization of the STIM monomers, followed by rotation of the multimers in the ER membrane [191]. The SOAR domain is released from CC1, the bundled STIM multimers spread out, and their C-terminal PBDs associate with the plasma membrane at the ER/PM junctions. Orai1 channels are drawn closer to the STIM, facilitating interaction of the N-terminal of Orai with the SOAR domain of the STIM. This promotes rotation of the Orai channel helices, which opens the channel pore for Ca^2+^ entry (Figure 8) [69,186,191].

There are many predicted/putative Ca^2+^/CaM-binding sites in the cytoplasmic domains of both STIM1 and STIM2 [199]. The lysine-rich PBD binds CaM with very high affinity in the presence of Ca^2+^ (Kd: 0.8 μM for STIM1 and 0.9 μM for STIM2). In its absence, the affinity is lower (Kd: 55 μM and 150 μM for STIM1 and STIM2, respectively) [200]. These levels indicate that cytosolic Ca^2+^ may modulate the STIM–CaM interaction. One proposal was that CaM might destabilise the ER–PM contacts at high Ca^2+^ concentrations by competing with phosphoinositides for binding to the PBD.

STIM1 has been shown to activate TRPC channels but through a mechanism not requiring the STIM1 N terminus or transmembrane domain. It was proposed that the C–C domains in the STIM1 C terminus bind to TRPC1, 4, and 5 and that the terminal PBD region gates the TRPC channels [201,202].

### 5.3. SOCE in the Placenta

In the Protein Atlas (https://www.proteinatlas.org/, accessed on 29 December 2024) seven entries reported expression of small amounts of mRNA of *Orai1* and eight of *STIM1*. These genes or their proteins were not highlighted in array studies of placental villi, syncytiotrophoblasts, or trophoblasts [24,25,26,100,164]. From this lack of data, it seems that Orai and STIM are not produced by the brush-border syncytium. This requires confirmation. If true, ER stores must be replenished with extracellular Ca^2+^ imported by TRP Ca^2+^ channels, possibly with a minor contribution by voltage-gated channels. *TRPC3*, *TRPC4*, and *TRPC6* mRNA and proteins were expressed in the syncytiotrophoblasts of human term placentas, and mRNA of these channels and of *TRPC1* and *TRPC6* was detected in first-trimester placentas [178].

Two studies demonstrated that pregnancy-specific beta-1-glycoproteins from the immunoglobulin superfamily, PSG1 [203] and PSG9 [204], increased production of Orai in foetal cell cultures. Both PSGs are secreted by syncytiotrophoblasts into the maternal circulation, and serum concentrations increase with gestational age. PSG1 may have a role in placental vascular development [205]. PSG9 has important roles in immune regulation, thrombosis regulation, and angiogenesis during pregnancy [204]. Compared with concentrations in healthy pregnant women, PSG1 levels were lower in the serum of individuals with early-onset pre-eclampsia (EOPE) [203,206], and serum PSG9 levels were significantly decreased in patients with pre-eclampsia [204].

PSG1 treatment of cultured human trophoblast HTR-8/SVneo cells upregulated the expression of Orai1 protein and phosphorylation of Akt triggered by Ca^2+^ entry through the Orai1 channel. The selective inhibitor of Orai1 (MRS1845) suppressed Akt signalling and decreased the migration of trophoblasts in response to PSG1. It was proposed that downregulated PSG1 may reduce the Orai1/Akt signalling pathway, thereby inhibiting trophoblast migration, and suggested that PSG1 may serve as a potential target for the treatment and diagnosis of EOPE. It is probable that this was a response of stromal cells to PSG1 [28]. HTR-8/SVneo cells are first-trimester extravillous trophoblasts, which were infected with simian virus 40 large T antigen (SV40). Cultured HTR-8/SVneo cells were shown to generate a predominance of stromal cells and relatively few trophoblasts, whereas choriocarcinoma cell lines (BeWo, JEG-3, and Jar) generated only trophoblasts [207]. Further investigation using trophoblast cell lines derived from human stem cells that develop into extravillous trophoblasts may help to resolve this issue [29].

PSG9 treatment (0.1 µg/mL) of human umbilical vein endothelial cells (HUVECs) significantly enhanced the expression levels of Orai1 and Orai2 and store-operated calcium entry (SOCE) and the expression of endothelial nitric oxide synthase (eNOS) and NO production. Pretreatment with an inhibitor of SOCE (3,5-bis(trifluoromethyl) pyrazole derivative, BTP2) abolished these responses to PSG9. eNOS is synthesised primarily by endothelial cells and regulates vascular tension by promoting NO synthesis. Ca^2+^ bound to calmodulin increases eNOS activity [208]. The study indicates that PSG9 may regulate the function of vascular endothelial cells by increasing Ca^2+^ entry via the Orai1 channel. The same mechanism may apply to PSG1. These findings may be relevant to pre-eclampsia, could have therapeutic potential, and should be pursued.

## 6. Ca^2+^ Clearance from the Cytosol

Excess Ca^2+^ may be removed from the cytosol by transfer to the ER, mitochondria, or extracellular space, as shown in Figure 9A.

### 6.1. Sarcoplasmic/Endoplasmic Reticulum Ca ^2+^-ATPase, (SERCA)

SERCA is a membrane transport protein found ubiquitously in the ER of all eukaryotic cells. Its major function is to transport Ca^2+^ from the cytosol into the ER to restore basal Ca^2+^ concentrations following oscillations to μM concentrations elicited by cell stimulation [209]. There are three members of the SERCA family (SERCA1–3). The genes are *ATP2a1/2/3*. mRNA of *ATP2a2* and *ATP2a3* was expressed in mouse trophoblasts at E14.5 pf., but *ATP2a1* was not detectable [102]. SERCA is expressed in oocytes [210,211,212]. However, only a fraction of Ca^2+^ from each Ca^2+^ oscillation is recycled into the ER Ca^2+^ pool by SERCA. A small amount of Ca^2+^ is transported into mitochondria by the electrophoretic pump [209], but most Ca^2+^ is extruded from the cytosol into the extracellular space via two plasma membrane transporters: plasma membrane calcium ATPase (PMCA) and Na^+^/Ca^2+^ exchanger (NCX).

### 6.2. Plasma Membrane Calcium ATPase (PMCA)

In mammals, four separate genes code for the major PMCA isoforms 1–4 (genes *Atp2b1–Atp2b4*) [213]. Differentiated trophoblasts from human term placenta expressed mRNA of *PMCA* 1–4 [214]. Mouse trophoblast homogenates expressed mRNA for *atpb1* (PMCA1) and *atpb4* (PMCA4) at E14.5 pf. but very little *atpb2* or *atpb3* [102]. Homozygous *atp2b1−/−* mice die in early embryonic life [215]. PMCA has 10 transmembrane domains, two large intracellular loops, and amino- and carboxy-terminal cytoplasmic tails. The carboxy tail contains the main regulatory sites for the activity of the pump: a CaM-binding domain, phosphorylation sites for protein kinases A (PKA) and C (PKC), and a high-affinity allosteric Ca^2+^-binding site. The carboxy terminal of full-length splice isoforms interacts with the PDZ domains of a variety of proteins, notably NHERF2 [209,213]. PMCA operates as a Ca^2+^:H^+^ exchanger with a 1:1 Ca^2+^/ATP stoichiometry. It has a high Ca^2+^ affinity, and many agents modulate its activity. The first loop contains sites for activation by phospholipids and autoinhibitory interaction with the CaM-binding domain. The second loop includes the binding sites for ATP, the acyl phosphate intermediate, and the second binding site for the carboxy-terminal CAM-binding domain [209].

The main regulator of PMCA function is Ca^2+^/CaM [216,217]. In the absence of CaM, the pumps are autoinhibited by the C-terminal tail, which binds to the two major intracellular loops. The release of autoinhibition requires binding of Ca^2+^/CaM to the PMCA C-terminal. This induces a conformational change in PMCA, with displacement of the C-terminal tail from the major catalytic domain [214,216]. Acidic phospholipids or phosphorylation of Ser/Thr residues in the C-terminal by PKC and/or PKA may facilitate these events. Inhibition of CaM prevents PMCA stimulation [106]. PMCA2b interacts with NHERF2 and may link the transporter to the underlying actin cytoskeleton and stabilise its local retention [218]. Whether PMCA1 interacts with NHERF1 is unknown.

### 6.3. Sodium–Calcium Exchangers (NCX)

Mammals express three NCX isoforms (NCX1, NXC2, and NXC3), the genes being *SLC8A1*, *SLC8A2*, and *SLC8A3*, respectively. They are low-affinity, high-capacity Na^+^/Ca^2+^ antiporters sited in the plasma membrane, with a central role in maintaining cellular calcium homeostasis for cell signalling. NCX1 is expressed widely, predominantly in the heart, kidney, and brain. NXC2 and NXC3 have a more restricted distribution [219,220,221].

NCX has a transmembrane (TM) domain with 10 TM helices and a large intracellular loop that separates the TM domain into two halves containing TMs 1–5 and TMs 6–10, respectively, and short extracellular N and intracellular C termini [219,222,223,224]. The TM domain mediates ion exchange, and the intracellular loop controls allosteric regulation by cytosolic Ca^2+^ and Na^+^ [219,225]. NCX catalyses the exchange of Na^+^ and Ca^2+^ with a 3:1 stoichiometry and is electrogenic [209,219]. The intracellular loop contains two Ca^2+^-binding domains (CBDs). CBD1 is the primary Ca^2+^ sensor. It detects small increases in cytosolic Ca^2+^ and undergoes large structural changes that activate the exchanger. CBD2 only binds Ca^2+^ at high concentrations and undergoes modest structural alterations [209]. Many other agents participate in regulation [220]. NCX1 is expressed in human placental tissue [226], and both NCX1and NCX2 isoforms in trophoblasts from human term placenta and in BeWo (choriocarcinoma) cells [214,227]. Under basal conditions, NCX does not have a major role in placental Ca^2+^ efflux [214].

## 7. Transcellular Calcium Transport to the Foetus

As the foetal organs grow and the skeleton develops, there is an increasing transcellular flow of Ca^2+^ across the placenta from the mother to the foetus, which rises exponentially in the third trimester. This must be accomplished without causing an increase in the syncytial cytosolic Ca^2+^ concentration and cell toxicity. It requires that Ca^2+^ is chaperoned throughout its passage through the placental villi and then rapidly discharged across the basal membrane to the extracellular space and foetal circulation. In contrast to the duodenum, Ca^2+^ is not absorbed additionally by the paracellular route [228,229]. In the duodenum, transcellular absorption is driven by 1,25(OH)_2_D and has been investigated intensively. Few data are available for the placenta.

### 7.1. The Working Model Proposed for Postnatal Intestinal Ca^2+^ Absorption

A working model [114,230] shows that Ca^2^ enters the microvillus through the TRPV6 calcium channel and is promptly taken up by CaM bound to brush-border myosin 1A (myo1A) [231]. Binding to myo1A may facilitate the movement of the Ca^2+^/CaM complex into the terminal web (shown in Figure 3). Here, Ca^2+^ transfers to calbindin-D_9K_ (CaBP-9k) [232,233], which has a higher affinity for Ca^2+^ than CaM [234], and is transported through the cytoplasm in endocytic vesicles. At the basolateral membrane, Ca^2+^ is discharged from the cell by the Ca-ATPase PMCA1b. In support of the proposed model, the following can be noted: (i) CaM is the major Ca^2+^-binding protein in the microvillus [234]. Inhibitors of CaM block 1,25(OH)_2_D-stimulated calcium uptake by brush-border membrane vesicles (BBMV) [235]. (ii) The 1,25(OH)_2_D increases the concentration of CaM in the microvilli. (iii) CaM/myo1A is demonstrable at the highest concentration in intestinal brush-border membrane cells with high capacity to transport Ca^2^ [236]. (iv) The 1,25(OH)_2_D enhances intestinal calcium transport by inducing the expression of *trpv6, cabp-9k*, and *pmca1b* [114,126,220,230,237]. (v) In vitamin-D-deficient animals, Ca^2+^ microanalysis showed Ca^2+^ accumulation along the inner surface of the plasma membrane of microvilli [230]. (vi) Following vitamin D or 1,25(OH)_2_D administration, Ca^2+^ is seen in mitochondria and vesicles within the terminal web [230]. (vii) Numerous tubules and vesicles containing Ca^2+^ were seen in the terminal web region of enterocytes from normal and vitamin-D-treated chicks. These were smaller and depleted in vitamin D deficiency [238].

Two CaBPs, calbindin-D_9K_ (CaBP-9k, gene *S100G*, locus Xp22.2) and calbindin-D_28K_ (gene *CALB1*, locus 8q21.3), act as cytosolic Ca^2+^ buffers to maintain low intracellular Ca^2+^ levels during changes in transcellular Ca^2+^ transport. Both bind Ca^2+^ with high affinity and are increased by 1,25(OH)_2_D [114]. Calbindin-D_9K_ is highly expressed in the small intestine and mediates Ca^2+^ transport [114,126,237]. In the small intestine, PMCa1b is the vitamin-D-regulated extrusion system and not NXC1 [230,239]. The deletion of *pmca1b* in mice reduces calcium absorption and growth and bone mineralization [240]. PMCA activation is dependent on CaM, which is its main regulator (Section 6.2) [114,216,240]. CaBP-9k may directly enhance PMCa1b activity [241]. Problems with the proposal are as follows: (i) Mice null for *s100g* have normal intestinal calcium transport, serum Ca^2+^ concentration, and bone mineralisation [242]. Another protein may act as a chaperone [243]. (ii) Ca^2+^ transport of myosin1-deficient mice is not reduced [244].

### 7.2. Transcellular Ca^2+^ Transport in the Placenta

The transcellular pathway for Ca^2+^ in the placenta similarly requires that Ca^2+^ is chaperoned in transit and is extruded rapidly. Data are limited. As in the duodenum, Ca^2+^ enters the villi of the human placenta via TRPV6 channels (Section 4), and PMCA1b has the principal role in extruding Ca^2+^, with a minimal contribution from NCX [214]. In rodents, *pmca* gene expression doubles during the last 7 days of gestation [233]. In humans, PMCA protein expression did not change during the third trimester of pregnancy, but the activity of the transporter increased linearly during this period [245], indicating activation by other agents such as CaM, PKC, and acidic phospholipids [246].

A major difference between placental and duodenal transcellular Ca^2+^ transport is that it is driven by oestrogens and not by 1,25(OH)_2_D [100]. In addition, there is uncertainty about whether CaBP-9k is the chaperone for placental transport in humans. There are species differences in calbindin expression, explained by complex regulation by sex steroids [247,248]. *s100g* was expressed in rat placenta and increased dramatically in late gestation [233]. Structural studies demonstrated a difference in two essential nucleotides in the oestrogen response element [ERE] between the rat and human genes. The human ERE failed to bind the oestrogen receptor. This could prevent CaBP-9k expression in the human uterus and possibly the placenta [247]. In the mouse uterus, *s100g* is mainly regulated by progesterone and not oestrogen, explained by a single-base difference in the mouse ERE compared with that of rats [248]. Nonetheless, mRNA and protein for CaBP-9k were expressed in human placental tissue collected at term [16]. RNA and protein of calbindin-D_28K_ were expressed in cytotrophoblasts and syncytiotrophoblasts from human term placenta [16,249], and calbindin-D_28K_ might potentially contribute to transport.


**Section C: Effects of Peptides and Hormones on Placental Ca**
**
^2+^
**
** Transport**


The provision of an adequate mineral supply to the foetus is critically dependent on a normally formed and functional villous system. The foundations for this are laid very early in embryonic life at the time of implantation. Numerous factors are required for this process and continued placental development. Two factors that disturb calcium transport in animal models with genetic deficiencies are insulin-like growth factor 2 (IGF2) and parathyroid hormone-related peptide (PTHrP).

## 8. Insulin-like Growth Factor 2 (IGF2)

IGF2 is a growth factor expressed in the placenta in many species, including humans and rodents. The *IGF2* gene has a complex genomic and transcriptional organisation with five promoters [3,250,251,252,253,254], which is normally expressed by only the paternal allele [250]. The P0 promoter operates specifically in the placenta, leading to high levels of *IGF2* expression in placental tissues during gestation. It is abundant in foetal tissues and circulation and in the maternal circulation [3,4,250,255,256]. IGF2 is expressed from very early in embryonic life and increases progressively to term and has been shown to be crucially involved in blastocyst implantation and for normal foetal and placental growth throughout pregnancy. Foetoplacental endothelial cells are a significant source [4]. Levels increased progressively in human cord blood from the 21st week of gestation to delivery, with the main increase after the 32nd week [255]. IGF2 binds with high affinity to insulin-like growth-factor-binding proteins (IGFBPs). A metalloproteinase, pregnancy-associated plasma protein-A2 (PAPP-A2), cleaves bioactive IGF from IGFBP-3 and -5 [256,257]. Free IGF2 binds principally to the type 1 IGF receptor (IGF1R) with high affinity [258]. This is a tyrosine kinase receptor widely expressed in the placenta, which activates mitogen-activated protein kinase (MAPK) and PI3 kinase signalling pathways [3,250,253,254,259,260]. IGF2 also binds with IGF receptor 2 (IGF2R), which has been shown to mediate IGF2 clearance through lysosomal degradation, and it can bind to the type A insulin receptor INSR-A [250]. In vitro, IGF2 stimulates the survival, proliferation, and differentiation of human placental trophoblasts [3,253] and, in mouse studies, the expansion of the placental vasculature to support foetal growth [4].

### Pathophysiology

Complete ablation of *Igf2* in mice *(Igf2* null) results in poor growth of the placenta, defective vascularization, increased barrier thickness and reduced surface area of the villous labyrinth, impaired amino acid transport capacity, and foetal growth restriction [261]. Deletion of the placental-specific *Igf2 PO* transcript in mice similarly causes placental growth restriction and compromised labyrinthine zone formation. However, adaptive upregulation of glucose, glutamine, system A amino acid, and Ca^2+^ transport [251] moderates the effects [254]. At embryonic day 17 (E17) pf., the foetal and placental weights of *P0* knockout (P0) mice were reduced when compared with WT (wild type) but the ratios of foetal to placental weight were significantly increased. Despite having similar rates of placental Ca^2+^ transport, P0 foetuses had a lower capacity for Ca^2+^ transport, and blood and body Ca^2+^ and bone mineralisation were reduced. By E19 pf., placental Ca^2+^ transport and bone mineralisation had increased, demonstrating an adaptive response [251]. Despite this, mRNA expression of placental Ca^2+^ pathway genes was generally reduced compared with WT [262].

The importance of IGF2 for foetal and placental growth is manifest in two human disorders [250]. Patients with mutations causing Silver–Russell syndrome have low levels of IGF2 expression, foetal growth restriction, and hypoplasia of the placenta and chorionic villi [263,264]. In contrast, individuals with Beckwith–Wiedemann syndrome with increased IGF2 due to bi-allelic *IGF2* expression caused by loss of imprinting have enlarged placentas and somatic overgrowth [265,266,267].

## 9. PTH and PTH-Related Peptide (PTHrP and PTHLH)

PTH-related peptide (PTHrP, alias PTH-like hormone, PTHLH, Online Mendelian Inheritance in Man (OMIM) entry 168470, gene *PTLH*) is a multifunctional protein that acts as a paracrine, autocrine, and intracrine factor [268,269,270] to regulate diverse physiological processes. These include cell proliferation and differentiation in bone and epithelial Ca^2+^ transport [271]. PTHrP is expressed from the very early stages of embryogenesis [268] by human and mouse trophoblasts and by immature chondrocytes in foetal bone and is abundant in the placenta and foetal membranes [268,272,273,274]. Human PTHrP is encoded by a single gene, namely, *PTHLH,* which undergoes complex translational and posttranslational processing. There are three principal secretory forms, namely, PTHrP (1–36), PTHrP (38–94), and PTHrP (107–139, osteostatin). PTHrP 1–36 shares homology with human PTH, with 8 of the first 13 residues being identical, and binds the peptide to the PTH receptor [275]. Hence, PTHrP can stimulate most of the actions of PTH. The peptide sequence from residue 14 has little similarity to PTH. Surprisingly, mid-region peptides, including amino acids 38–94, 38–95, and 38–101, stimulate placental Ca^2+^ transport and not the N-terminal as might be predicted [276,277]. A lysine/arginine-rich sequence in the mid-region may directly import PTHrP into mitochondria via importin β/Ran GTPase, enabling intracrine signalling [274,278]. The C-terminal fragment, namely, residues 107–139, has actions on skin, heart, and bone cells [274].

### 9.1. The PTH/PTHrP Receptor (PTH1R)

The PTH/PTHrP receptor, PTHR1, is also expressed in the placenta. It is a class B1 G-protein-coupled receptor (GPCR) [41]. There are three domains: a large extracellular N-terminal, ligand-binding domain (ECD) [279,280,281]; a transmembrane domain with seven helices linked by three extracellular loops and three intracellular loops [279]; and an intracellular C-terminal [279,280,281]. PTH1R signals primarily via Gs, which stimulates adenylyl cyclase activity but can also couple to Gq/11, which activates phospholipase C (PLC), G12/13, and Gi/o. The Gγ subunit can couple to diverse transducer proteins. PTH 1–34 and PTHrP 1–36 molecular fragments both bind to the ECD of the PTHR. However, PTH fits more tightly into the receptor’s binding cleft [275]. Binding of the PTH or PTHrP peptides to the ECD can bias signalling [279,280]. One important manifestation of this is that while both PTH and PTHrP signal from the cell surface, PTH can also induce prolonged signalling from early endosomes [115]. The PTH1R has two distinct active conformations (RG and R0). The RG conformation is G protein dependent and is associated with transient cAMP responses from the plasma membrane. It is stabilised by PTH1–34 and PTHrP1–36 indistinguishably. The R0 state is stabilised preferentially by PTH and determined by PTH activation of Gq and formation of a Gβγ, PTH/PTH1R, β-arrestin complex. This is internalised and promotes endosomal signalling [282].

### 9.2. Roles in the Embryo and Foetus

PTHrP is detected in mouse embryonic and extraembryonic tissues from the late morula stage onwards [268,283] and is required for blastocyst formation [284]. Furthermore, it acts as a potent vasodilator of the foetoplacental vasculature to maintain low-resistance blood flow [285]. PTHrP has essential roles in regulating normal placental morphogenesis and functional development, attainment of normal foetal growth, regulating foetal Ca^2+^ homeostasis and placental Ca^2+^ transport [286,287], and skeletal development [286,288,289]. PTHrP is increased in cord blood in foetal growth restriction [262]. The mechanism is unknown but may be an adaptive response of the vascular endothelium to hypoxia and oxidative stress to increase nitric oxide (NO) and blood vessel dilatation as has been suggested for pregnancy-specific beta-1-glycoprotein 9 (refer to Discussion). In a study of 78 pregnancies complicated by gestational diabetes, immunochemical analysis demonstrated PTHrP and PTHR1 expression in extravillous cytotrophoblast and in the uterine decidua, with small amounts in the syncytiotrophoblast and villous cytotrophoblast and none in the villous stroma. Placental PTHrP and PTHR1 expression in the extravillous cytotrophoblast was associated with a higher incidence of maternal abnormal fasting glycemia in an oral glucose tolerance test at 24–28 weeks of gestation (indicating insulin resistance), compared with a normal fasting but abnormal 60′ or 120′ glycaemia (indicating insulinopaenia). PTHR1 expression in the extravillous cytotrophoblast was associated with a lower foetal weight in the third trimester estimated by ultrasound scanning and a lower foetoplacental weight ratio. PTH-rP expression in the syncytiotrophoblast was associated with higher incidence of maternal obesity and neonatal Apgar score at 1 min <7 [290].

*Pthlp*-null mice die in mid-gestation [288,289], and mice expressing a truncated form of PTHrP die in the early postnatal period [291]. The placental weight of PTHrP-null mutants was reduced and associated with a reduced foetal to placental weight ratio [291]. The placental labyrinths were fissured, and the cells were disorganised and showed increased apoptosis [268]. Disrupted cell–matrix interactions resulting from changes in the profile of integrins expressed by PTHrP may have been contributory [292,293]. The blood Ca^2+^ concentration of null mice was significantly reduced but was corrected by foetal injection of hPTHrP (1–86) and hPTHrP (67–86) [287]. Placental Ca^2+^ transport and Ca^2+^ accretion were increased, indicating adaptation to Ca^2+^ depletion, but expression of *trpv6* and *PMCA isoforms 1* and *4* were unaltered [286].

## 10. The Calcium-Sensing Receptor (CaSR)

The CaSR is a class C GPCR that couples to multiple G-protein subtypes to activate intracellular signalling pathways [294]. It is expressed in the parathyroid glands [295], as well as by chondrocytes and osteoblasts in bone, with its highest expression seen in the hypertrophic chondrocytes [296]. In postnatal life, the CaSR in the parathyroid glands is activated by an increased serum Ca^2+^ concentration and suppresses PTH secretion. This in turn leads to decreases in renal 1,25(OH)_2_D_3_ synthesis and in bone turnover. In the foetus, the CaSR is probably activated by the raised serum Ca^2+^ and suppresses PTH secretion, accounting for the low observed serum PTH levels [1]. The CaSR promotes chondrocyte differentiation in foetal bone [296].

The CaSR primarily exists at cell surfaces as a disulphide-linked homodimer. The extracellular domain comprises a bi-lobed Venus flytrap domain and a cysteine-rich domain [297,298]. In the inactive state, the two CaSR protomers interact primarily at the lobe 1–lobe interface. They rotate on activation, and this extends the interface between them [297]. The ligand-bound CaSR receptor predominantly uses the G_i/o_ pathway to suppress cAMP and activate MAPK signalling cascades and the G_q/11_ pathway, which activates Ca^2+^ mobilisation and MAPK signalling [294,295].

### 10.1. Mutations of the CaSR in Mice

In mice, ablation of one (*Casr*+/−) or both (*Casr* null) alleles of *CaSR* increased the foetal blood Ca^2+^ above the WT value and increased circulating PTH and 1,25(OH)_2_D_3_ [299]. The placenta might have been the source of the increased 1,25(OH)_2_D_3_ [100]. The circulating plasma PTHrP was significantly lower in *Casr*-null foetal mice compared with WT siblings, perhaps indicating the effects of the CaSR to downregulate PTHrP expression [300]. Placental transport of ^45^Ca was significantly, and dose-dependently, reduced in *Casr+*/− and *Casr*-null foetuses compared with WT siblings [299]. The explanation is unknown. One possibility might be that TRPV6 and/or other Ca ^2+^ entry channels in the placental villi were inactivated by the increased Ca ^2+^ (Section 4). Bone resorption was increased in global *Casr*-null foetuses in association with increased circulating PTH levels, increased excretion of Ca^2+^ and the bone resorption marker deoxypyridinoline into amniotic fluid, and a significant reduction in the mineral content of the foetal skeleton [1,299]. Chondrocyte-specific ablation of *Casr* (*CasrChon*-null mice) caused severely delayed cartilage and bone development and embryonic death [295,296].

### 10.2. Mutations of the CaSR in Humans

More than 400 different germline loss- and gain-of-function *CaSR* mutations giving rise to disordered Ca^2+^ homeostasis were identified by 2019 [294]. The mutations may be inherited or occur de novo. *CaSR*-inactivating mutations cause neonatal severe hyperparathyroidism (NSHPT). The median age of the diagnosis of NSHPT is 14 days, ranging from 2 days to several months of age. No case reports have described blood Ca^2+^ or PTH values in foetuses or newborns [1]. Presentation is with symptoms of hypercalcaemia and skeletal demineralization with fractures. The parathyroid glands are grossly enlarged, and serum PTH is high. Without urgent parathyroidectomy, this condition is almost invariably fatal [294,301]. Most reported cases are due to homozygous (around 75%) or compound heterozygous mutations (around 10%). A minority of patients had heterozygous mutations of *CaSR* located in regions that are critical for receptor activation [297,298]. Individuals heterozygous for dominant *CaSR* mutations (familial hypocalciuric hypercalcaemia) have mild or moderately raised serum Ca^2+^, which is often asymptomatic and detected as an incidental finding in later life [295] and has not been associated with foetal Ca^2+^ disturbances. Mutations in the Gα11 protein and the adaptor protein-2 sigma subunit (AP2σ), by which the CaSR is internalised, cause CaSR signalling disturbances, demonstrating novel mechanisms by which CaSR is internalised and that CaSR can signal by an endosomal pathway [302].

## 11. Calcitonin and Calcitonin Gene-Related Peptide (CGRP)

Identifying the roles of calcitonin in the placenta is proving difficult for four reasons: (i) the calcitonin gene (*CALCA*, alias *Ctcgrp1*) encodes two distinct proteins, calcitonin and calcitonin gene-related peptide (CGRP), with multiple isoforms produced by gene splicing; (ii) the two proteins bind to different receptors, CALCR and CRLR; (iii) the gene is expressed by both the uterine endometrial decidua and the foetal trophoblast; and (iv) the nomenclature is confusing. The gene has six exons, which are all included in the primary RNA transcript. *Calcitonin* or *CGRP* mRNA is formed subsequently by posttranscriptional processing. Transcription of *CGRP* is downregulated by vitamin D. A second CGRP homolog, β-CGRP, is produced by a separate gene [303,304,305,306,307]. Human CGRP and β-CGRP differ by three amino acids [308].

### 11.1. Calcitonin

Calcitonin is a 32-amino-acid peptide calcium-lowering hormone. It is synthesised in the parafollicular cells or “C cells” of the thyroid gland [304] and is liberated by endocrine secretion. It regulates Ca^2+^ levels in bone and kidney cells. The placenta has been shown to express calcitonin mRNA and protein and the calcitonin receptor [309]. Calcitonin may have a critical role during blastocyst implantation. There was a sharp burst of calcitonin mRNA and protein expression in the endometrium of pregnant rats from day 2 to day 5 pf., encompassing the time of blastocyst implantation, which ended abruptly by day 6 pf., when implantation was completed [310,311]. Expression was localised to the endometrial glandular epithelial cells and stimulated by progesterone. Oestrogen in a low dosage synergised with progesterone but had no effect alone [310,311]. Attenuation of calcitonin expression blocked the implantation of the early embryo [311]. There are no reports of human genetic calcitonin deficiency.

### 11.2. The Calcitonin Receptor (CALCR, CTR)

The calcitonin receptor is a class B GPCR. Numerous alternatively spliced transcripts of the human *CT* receptor gene (*CALCR* alias *CTR*) have been reported. In contrast to the CGRP receptor (Section 11.3), the calcitonin receptor does not require receptor-activity-modifying protein (RAMP1) to bind and respond to calcitonin. Two splice variants of the human CT receptor differ by the presence (CT_(b)_) or absence (CT_(a)_) of 16 amino acids in the first intracellular loop. The CT_(a)_ receptor is expressed widely. In most tissues, CT_(b)_ receptor expression is low [312] but is reported to be significant in the ovary and placenta [305]. The two variants have similar affinity for peptide ligands, but unlike the CT_(a)_ receptor, the CT_(b)_ receptor is poorly internalised and has altered coupling to the G protein in response to stimulation, with no Gq-mediated responses and attenuated Gs-mediated signalling [305]. *CALCR* ablation causes embryonic death at mid-gestation for unknown reasons [1].

### 11.3. CGRP, the Calcitonin Receptor-like Receptor (CRLR), and the Functional CGRP Receptor

Published data for foetal expression of *CGRP* are scant. In three very premature foetuses, *CGRP* was expressed in the endothelium of human umbilical cord blood vessels at delivery. The % methylation of the promoter of *CGRP* was higher in cord blood DNA from preterm than term infants [313]. Progesterone stimulated *CGRP* expression, while oestrogen inhibited it [314]. CGRP is a potent vasodilator [315]. It was suggested that CGRP may have a role in increasing blood flow through the foetoplacental unit in late gestation [314].

The functional CGPR receptor is a complex of CGRP, the calcitonin receptor-like receptor (CRLR), RAMP1, and an additional intracellular protein, namely, the receptor component protein (RCP) [303,305,307,316]. The CRLR by itself is not expressed significantly at the cell surface and does not respond to any known ligand. With RAMP1 it becomes the CGRP receptor (i.e., CRLR/RAMP1). RAMP1 is an intrinsic protein with an extracellular N terminus, a single transmembrane domain, and a short intracellular domain [305,307,317]. Immunoreactive CGRP receptors were demonstrated in the placental labyrinth of foetal rats in the trophoblasts and in the endothelium and underlying smooth muscle cells of villus blood vessels. Levels increased 10-fold from day 17 pf. to a peak before labour on day 22 pf. [314].

### 11.4. Calcitonin and CGRP in Regulation of Mineral Status

A small number of *calca (ctgrp)-null mice* died. The survivors had normal lifespans and were fertile. The serum Ca^2+^, placental transport of Ca^2+^, and skeletal Ca^2+^ content were not significantly different from littermate controls [318], demonstrating clearly that foetal-placental Ca^2+^ transfer does not require calcitonin or CGRP. Notable observations were that Mg^2+^ was significantly reduced in serum of *calca*-null mothers and foetuses, as well as in the foetal skeleton. There was no obvious explanation [318]. One possibility might be that calcitonin or CGRP interacts with the TRP melastatin cation transporters TRPM7 and/or TRPM6, which interact and are both expressed by placental trophoblasts (Section 4.1). TRPM7 forms homotetrameric channels that are highly permeable to Ca^2+^, Mg^2+^, and Zn^2^ and are regulated by Mg^2+^. Depletion of intracellular Mg^2+^ and Mg-ATP promotes TRPM7-mediated uptake of extracellular Mg^2+^ [319,320]. The interaction of TRPM6 with TRPM7 was proposed to increase epithelial Mg^2+^ transport [321].

## 12. Vitamin D and Placental Ca^2+^ Transport

### 12.1. Importation of 25OHD and Activation to 1,25(OH)2D

Serum ionised Ca^2+^ concentrations in human cord blood are uniformly around 0.30–0.50 mM higher than in maternal serum [1]. A summary of data for serum vitamin D metabolites in foetal rodents and lambs showed that circulating concentrations of 1,25-dihydroxyvitamin D_3_ (1,25(OH)_2_D_3_) were low and that in foetal mice, 24,25-dihydroxyvitamin D_3_ (24,25(OH)_2_D_3_) was 54-fold higher than 1,25(OH)_2_D_3_ and three times higher than maternal 24,25(OH)_2_D_3_ [322]. Published data for human foetuses are sparse. In one study, cord sera from 21 preterm foetuses were compared with paired maternal sera. The cord and maternal serum Ca^2+^, Mg^2+^, and phosphorus concentration correlated closely, but levels of all three analytes were significantly higher in the cord samples. Cord serum 25-hydroxyvitamin D_3_ (25(OH)D_3_), 24,25(OH)_2_D_3_, and 1,25(OH)_2_D_3_ levels were significantly lower than those observed for the mothers. Whereas the concentrations of 25(OH)D_3_) and 24,25(OH)_2_D_3_ in cord blood both correlated directly with maternal blood levels, 1,25(OH)_2_D_3_ levels showed no association. Maternal 1,25(OH)_2_D_3_ levels were not related to the gestational age or to maternal Ca^2+^, Mg^2^, inorganic phosphate, or 25(OH)D_3_ concentrations. Cord 1,25(OH)_2_D_3_ correlated significantly only with cord Ca^2+^ levels [323]. Other studies found that (i) the average ratio of foetal serum 25(OH)D_3_ at 39.5 ± 1.2 weeks of gestation and maternal serum at 34–35 weeks of gestation was 0.67 (interquartile range, 0.56–0.84), with a correlation of *r* = 0.89 (*p* < 0.0001) [324], and (ii) cord blood concentrations of 25(OH)D_3_ and 1,25(OH)_2_D were tightly correlated (*r* = 0.78, *p* = 0.0001), in contrast to adult blood, in which 1,25(OH)_2_D remained relatively constant over a range of 25(OH)D concentrations (*r* = 0.02, *p* = 0.89) [325].

More than 99% of serum vitamin D metabolites are bound to carrier proteins, 85–88% with high affinity to D-binding protein (DBP) and 12–15% to albumin [326,327]. In the kidneys, both DBP and albumin with their vitamin D ligands bind to megalin and are internalised into apical endosomes by a clathrin-dependent endocytosis [100,328,329,330]. They are then transported via the endosome/lysosome pathway to lysosomes, where the binding proteins are degraded, and the ligands are released [327]. The 25(OH)_2_D_3_ is hydroxylated to inactive 24,25(OH)_2_D_3_ by CYP24A1 or taken into mitochondria for hydroxylation to 1,25(OH)_2_D_3_ by *CYP27B1*. It has now been shown that 25(OH)_2_D_3_ is taken up actively from the blood by trophoblasts from human term placentas into sub-membranous vesicles, probably by clathrin-dependent endocytosis [100]. The trophoblast cell surface is actively endocytic and has numerous coated pits and vesicles located between the microvilli and in the apical cytoplasm [331,332,333]. Placental syncytial trophoblasts express megalin [334]. The 1α-hydroxylase (*CYP27B1*) and 24-hydroxylase (*CYP24A1)* are both expressed in syncytiotrophoblasts in human placenta [100]. The mechanism by which 25(OH)_2_D_3_ reaches the mitochondria for activation to 1,25(OH)_2_D_3_ is uncertain. Intracellular vitamin-D-binding proteins (IDBPs) from the heat shock protein 70 family have been identified [327]. IDBP was found to bind 25(OH)_2_D_3_ and other steroid hormones. One proposal is that IDBP may chaperone 25(OH)_2_D_3_ for delivery to the mitochondria for activation [335]. An alternative proposal is that megalin may shuttle 25(OH)_2_D_3_ with its binding protein to the mitochondria via the retrograde early endosome to the Golgi pathway [330].

### 12.2. From Generation of 1,25(OH)_2_D_3_ to Activation of Gene Translation

The 1,25(OH)_2_D_3_ binds to the vitamin D receptor (VDR) with a very high affinity of 0.1 nM [336]. Ligand-bound VDR heterodimerises with the nuclear receptor retinoid X receptor (RXR). This increases the affinity of the VDR for the vitamin D response element (VDRE) in the promoter region of vitamin-D-responsive genes. A complex of supportive nuclear proteins is recruited to the site. These include possible co-receptors, pioneer factors, co-factors, members of the Mediator complex [337], chromatin modifiers, and chromatin remodellers [336]. Corepressors typically work by recruiting histone deacetylases (HDACs) or methyl transferases (MTs) to the gene. These coregulators can be specific for different genes and may be expressed differentially in different cells, thus providing some specificity for the actions of 1,25(OH)_2_D_3_/VDR. In addition, 1,25(OH)_2_D_3_ can affect the epigenome via direct interaction of the VDR/RXR complex with chromatin-modifying enzymes or indirectly by regulating genes that encode chromatin modifiers [336].

### 12.3. 1,25(OH)_2_D/VDR in Ca^2+^ Regulation in Pregnancy

The 1,25(OH)_2_D_3_ has a central role in maintaining normal Ca^2+^ status postnatally by promoting transcellular and paracellular Ca^2+^ absorption in the duodenum and kidneys. This is largely through its actions to coordinate transcription of the numerous proteins involved in operating the Ca^2+^ transport pathways. This is not the case for placental transport, which is driven by oestrogens and not by vitamin D. A review of evidence from numerous sources found that vitamin D, 1,25(OH)_2_D_3_, and the VDR are not required for the foetus to maintain normal serum mineral concentrations [1]. This conclusion is substantiated by observations of inherited disorders of vitamin D metabolism in humans that are not manifest at birth but present later in infancy. Individuals with inherited deficiencies of *CYP27B1* or of the VDR appear normal at birth and generally present with hypocalcaemia and/or rickets at 2 to 24 months [338,339] and 2–8 months of life [338], respectively. Infants with biallelic loss-of-function mutations in *CYP24A1* (idiopathic infantile hypercalcemia) presented with severe hypercalcemia at 6 to 8 months of age [340].

A small study found increased expression of *hOGG1* (human 8-oxoguanine DNA glycosylase) mRNA in placentas from pregnancies with pre-eclampsia (*n* = 20) or gestational diabetes (GDM, *n* = 20) indicative of oxidative stress, compared with 20 controls. The bone mineral content (BMC) of newborns was reduced. Both groups had lower concentrations of cord blood 25(OH)_2_D_3_; significantly lower placental mRNA expression of *CYP2R1* (*cytochrome P4502R,* product vitamin D 25-hydroxylase), *VDR*, *TRPV6*, *TRPV5*, *CABP9k*, *CaBP28k PMCA1,2,3*, *IP3R*, and the ryanodine receptors *RyR1,2,3*; and significantly higher expression of *CYP27B1* and *CYP24A.* There was a negative correlation between BMC and *CYP2R1*, *CYP24A1*, *VDR*, *CABP28K*, and *PMCA2*. It was proposed that hypoxia disturbed vitamin D metabolism and calcium transport [341]. Another study, similarly, found that compared with controls (*n* = 40), serum 25(OH)_2_D_3_ was lower in maternal and cord blood from 41 pregnancies with gestational GDM. Placenta and umbilical cord tissues from GDM pregnancies had significantly higher mRNA and protein expression of CYP24A1, but unlike the above study, the expression of VDR was higher, and CYP27B1 was significantly lower [342]. Larger studies are required to establish whether vitamin D metabolism is disturbed in GDM and pre-eclampsia and to investigate the functional significance if confirmed.

## 13. Genetic Disorders of Placental Ca^2+^ Transport

Observations of inherited defects demonstrate the roles of Ca^2+^ trafficking in vivo. Table 2 lists genetic disorders of calcium transport that are manifest in the human foetus in utero and/or at delivery.

Other disorders of the calcium supply chain present with a variety of clinical problems from early infancy to adult life [356]. Biallelic loss of *Orai1* causes severe combined immunodeficiency (SCID) presenting with recurrent infections and autoimmunity from a few weeks of birth [189,357,358]. Heterozygous autosomal dominant mutations of *Orai1* or STIM1 cause an overlapping spectrum of disorders of skeletal muscle contraction and platelet function [359], which include tubular aggregate myopathy (TAM) [189,357], Stormarken syndrome [360], and York platelet syndrome [189,357]. Deficiency of calsequestrin, an ER Ca^2+^-binding protein, causes myopathy [361]. Inherited deficiencies of *CYP27B1,* VDR, and CYP24a1 generally present after 2 months of age (Section 12) [338,339,340]. TRPC6 deficiency causes renal focal glomerulosclerosis in childhood [362]. Inherited syndromic defects of other TRPCs, SERCA, IP3/IP3R, or calbindin D9k have not been reported.


**Section D: The Developmental Origins of Health and Disease (DOHaD) and Postnatal Bone Development**


## 14. DOHaD and Postnatal Bone Development

David Barker [363] first suggested that adverse nutritional and environmental exposures during pregnancy may programme a foetus to have a higher risk for common chronic diseases in adult life. This concept of developmental plasticity, now termed the developmental origins of health and disease (DOHaD), has gained momentum and stimulated extensive research [2,17,364,365]. Epidemiologic observations that smaller size or relative thinness at birth and during infancy is associated with increased rates of coronary heart disease, stroke, type 2 diabetes mellitus, adiposity, metabolic syndrome, and osteoporosis [366] in adult life have been extensively replicated [17]. Developmental plasticity requires stable modulation of gene expression, and this appears to be mediated, at least in part, by epigenetic processes such as DNA methylation and histone modification [17,367]. Such modifications may subsequently promote chronic diseases under favourable enriched environmental conditions [365].

### 14.1. Evidence for an Association of Intrauterine Factors with Bone Health in Later Life

#### 14.1.1. Observational Epidemiologic Studies

One small study found that a low level of maternal serum 25(OH)D during pregnancy was associated with a small, significant reduction in bone calcium content at 9 years of age, which was not evident at birth [368]. No association was observed in a larger study [369]. High maternal intakes of Ca^2+^ and folate [370] and of phosphorus and protein, as well as maternal serum homocysteine and vitamin B12 concentrations in the first trimester [371], were associated with an increase in bone mineral at 6 years of age. Roseboom et al. (2019) [365] and Vaiserman and Lushchak (2021) [372] reviewed studies of health outcomes of in utero famine exposure [373,374,375] and findings from birth cohorts from Hertfordshire, UK; South Africa; Finland; the United States; and Brazil [365]. None of these studies appear to have addressed effects on postnatal bone calcification. However, prenatal malnutrition in Dutch [376,377] and Gambian [375,378] famine exposure was associated with alterations in methylation patterns in adult life, which correlated with phenotypic outcomes. This suggested that early exposure to nutrient restriction may influence epigenetic control of metabolic processes later in life. Observations from two studies of a population-based mother–offspring cohort [379,380] provided evidence for this. One study found a significant negative correlation between the percent methylation of RXRA at five of six CpG sites analysed in cord blood with bone mineral content at 4 years of age [379]. The other study demonstrated a significant inverse association of methylation at sites in the promoter of *CDKN2A*, which encodes long non-coding RNA ANRIL in cord blood with bone mineralisation at 4 and 6 years of age [380]. Differential DNA methylation of this gene at birth was shown previously to correlate with childhood adiposity [381].

#### 14.1.2. Intervention Studies in Which Women Received a Vitamin D Supplement

Two prospective controlled studies of children in the first 6–7 years of life indicate that high doses of antenatal vitamin D supplementation have beneficial effects on offspring skeletal mineralisation. In the Maternal Vitamin D Osteoporosis Study (MAVIDOS), 1000 IU of cholecalciferol was prescribed daily from 14 weeks of gestation to delivery [382,383]. There was no difference from placebo-treated controls at birth in dual-energy X-ray absorptiometry (DXA) bone scans. At 4 years of age a small increase in whole-body-less-head (WBLH) bone mineral density (BMD) was observed. Scans of 447 children followed up at ages 6–7 y demonstrated that offspring of gestationally supplemented mothers had higher WBLH bone mineral content (BMC) [0.15 SD, 95% confidence interval (CI): 0.04, 0.26], bone mineral density (BMD) (0.18 SD, 95% CI: 0.06, 0.31), and bone mineral apparent density (BMAD) (0.18 SD, 95% CI: 0.04, 0.32) compared with the controls. In the Copenhagen Prospective Studies on Asthma in Childhood 2010 (COPSAC2010) trial [384], high-dose vitamin D supplementation with 2800 IU/d from pregnancy week 24 to 1 week postpartum was compared with a standard dose of 400 IU/d. A combined analysis of DXA scans at age 3 and 6 years of 517 children showed that children in the vitamin D vs. placebo group had higher whole-body BMC, where the mean difference adjusted (aMD) for age, sex, height, and weight was 11.5 g (95% CI, 2.3–20.7; *p* = 0.01); higher WBLH BMC aMD, 7.5 g (95% CI, 1.5–13.5; *p* = 0.01); and higher head BMD aMD, 0.023 g/cm^2^ (95% CI, 0.003–0.004; *p* = 0.03). The largest effect was in children from vitamin-D-insufficient mothers and among winter births. There was a small decrease in the incidence of fractures in the vitamin D group (*n* = 23 vs. *n* = 36; incidence rate ratio, 0.62 (95% CI, 0.37–1.05; *p* = 0.08)). A review of 10 randomised and quasi-randomised placebo-controlled trials evaluated the effect of vitamin D supplementation alone or combined with calcium or other micronutrients during pregnancy [385]. None addressed foetal calcium and bone status. The evidence of benefit for pre-eclampsia, preterm birth, and birthweight was uncertain.

### 14.2. Epigenetic Processes and DOHaD

Epigenetic modifications of DNA in the foetus in response to adverse conditions in utero are proposed to contribute to poor metabolic health later in life [2,17,79,364,386,387,388]. Epigenetic activities modulate gene expression in numerous signalling pathways controlling key functions of the human placenta. Mechanisms include DNA methylation, histone tail posttranslational modifications (PTMs), and non-coding RNAs (ncRNAs). DNA methylation and histone PTMs regulate chromatin accessibility to transcription factors to facilitate or repress gene expression. ncRNAs are strong posttranscriptional regulators (reviewed [389,390]). DNA is methylated by covalent modification of carbon-5 of cytosine, catalysed by DNA methyltransferases (DMNTs). Methylation is erased by ten-eleven translocases (TETs) [391,392,393,394,395,396]. DNA methylation patterns in 12 cancer cell types showed that genes regulating Ca^2+^ transport are major targets of hypermethylation and downregulation [397]. In mice, knock-down of *tet2* decreased muscle stem cell proliferation and differentiation and disrupted calcium homeostasis. Muscle cell Ca^2+^ concentration was lower in *tet2*-null cells than in wild-type cells, and RNA expression of calcium-pathway-related genes was drastically reduced. Hypermethylated genes were predominantly enriched in the calcium pathway [396].

A critical risk period for introducing persisting disturbances of DNA methylation must be in very early embryonic life at around the time the blastocyst is implanted in the uterine decidua and embryogenesis is commencing [79]. Prior to fertilisation, genes of male and female gametes are selectively methylated. Starting immediately after fertilisation, both paternal and maternal DNA is demethylated via TETs. Simultaneously, phospholipase C PLCζ is activated, resulting in the release of Ca^2+^ from the ER stores and Ca^2+^ oscillations, which persist for about 2–4 h postgamete fusion [79]. Demethylation is essential to abolish gamete identity and confer cells of the growing embryo with pluripotent potential [79,392,394,398,399]. DNA methylation is regained following implantation and proceeds in a lineage-specific pattern. Histone-posttranslational modifications and epigenetic modifications are also reprogrammed during early embryogenesis [400]. Abnormal postfertilization Ca^2+^ profiles disrupt gene expression [79,401]. Dampened Ca^2+^ oscillations jeopardise preimplantation development, with RNA processing and polymerase II transcription genes particularly affected [402]. Hyperstimulation of Ca^2+^ oscillations compromises postimplantation development. These observations indicate that interventions to reduce the risk of adverse epigenetic changes should probably be well established before pregnancy.

## 15. Discussion

It has not been too long since it was commonly thought and taught that the main requirement for Ca^2+^ in utero is to mineralise the foetal skeleton and that to meet this need, Ca^2+^ is transferred from maternal blood to the foetus by 1,25(OH)_2_D_3_ synthesised in the placenta. The mechanisms were unclear but were probably, like intestinal Ca^2+^ absorption, promoted by circulating 1,25(OH)_2_D_3_. Over the last two or three decades, we have learnt that Ca^2+^ transport in the foetoplacental unit is far more complex than could have been envisaged. This is because of the power of the Ca^2+^ ion to orchestrate the activity of numerous signalling pathways and to direct/change cellular processes within milliseconds [67]. This additional role must be tightly regulated and matched to need and requires the interaction of numerous proteins, carriers, and transporters. The processes involved are under intensive investigation on a broad front. The aim here was to pull the data together to obtain an overview of our current understanding of the mechanisms and to highlight areas needing more clarification.

A normally formed and functional placental villous system with a large covering of terminal villi—the workforce of the mature placenta—is of paramount importance for providing an adequate mineral supply to the foetus. This was demonstrable for two proteins, IGF2 (Section 8) and PTHrP (Section 9), for which a deficiency in rodent models was associated with abnormal placental structure and reduced Ca^2+^ delivery in late gestation. Both were expressed very early in embryonic life, at the critical time for blastocyst implantation. In humans with Silver–Russell syndrome with low levels of IGF2 expression, the placenta and chorionic villi are hypoplastic, and foetal growth is restricted (Section 8 and Section 13). It is important to note that *IGF2*, *PTHrP*, and TRPV6 are expressed in bone as well as the placenta and appear to operate independently at these sites. How much the local effects of deficiencies of these proteins in bone contribute to the skeletal changes in global defects is unknown.

The placental villi must be equipped with a variety of Ca^2+^-importing channels, which can collectively respond to a range of different stimuli. There is abundant evidence that TRPV6 is the principal Ca^2+^ channel of the terminal microvilli, which are operative from around 16 weeks of gestation (Section 4.2). TRPV6 expression rises in synchrony with expansion of the placental syncytium and increases exponentially in the third trimester. Identification of human inherited TRPV6 deficiency in 2017 was a clear demonstration of its physiological importance. The expression of TRPV6 is low at earlier stages in gestation when the requirement for bones is also low. Other channels from the TRP family expressed in the placenta probably meet the Ca^2+^ requirement for signalling, despite their lower Ca^2+^ selectivity (Section 4.1).

What is not clear is how channels embedded in the apical membranes of the terminal microvilli are transported in and out of the villi and secured in the membranes. Like duodenal microvilli, they have a central actin core, but the transport mechanisms are enigmatic. The scaffolding protein EBP50, renamed NHERF1, was first isolated from placental villi (refer to Section 2: Terminal villi), but this appears to have been largely ignored, and its role in the placenta has not been pursued. In the microvilli of the renal proximal tubules, NHERF1 is an extremely versatile scaffold located close to the villus membranes. It provides support for transporters at the cell surface; links them to the cell cytoskeleton; and, importantly, connects them with large transiently assembled signalling complexes. In the kidneys, NHERF1 has an important role in the regulation of the sodium phosphate transporters NaPi-2a and NaPi-2c and the operation of the Na^+^/H^+^ exchanger NHE3 [115]. These proteins are all active in placental transport, and it seems likely that NHERF1 similarly has a role in the function of placental transporters. This merits investigation, as does the role of ezrin, which accounts for 5% of proteins in villus extracts and is another “neglected” placental protein (Section 2). Ezrin is important for linking membrane proteins with the actin cytoskeleton and may have a role in the transport of villus membrane channels and receptors, and possibly of the Ca^2+^ bound to CaM in the microvilli.

Ca^2+^ absorbed by the placental villi for export to the foetus, and not for local use in the placenta, must be chaperoned during transport through the cytosol to the basal membrane for extrusion by Ca^2+^-ATPase. In duodenal villi, calbindin D9k (CaBP-9k) is the principal carrier, and its expression is increased by 1,25(OH)_2_D_3_. The identity of the chaperone in the placenta is uncertain, and data are lacking. Placental Ca^2+^ absorption is driven by oestrogens and not by 1,25(OH)_2_D_3_. There are differences between species in placental expression of CaBP-9k, which may be attributable to small differences in the nucleotide sequence of the oestrogen response element on the CaBP-9k gene *s100g.* It is not known for certain whether the human placenta expresses CaBP-9k. This is another issue that requires clarification.

Cells maintain a store of Ca^2+^ in the ER, which is immediately available for signalling when membrane receptors are stimulated. Clearly, when the stores are getting low, they must be topped up with Ca^2+^ brought into the cells from the extracellular space. In many tissues, this is achieved by a partnership between a protein located in the ER, STIM1, and a Ca^2+^ channel, Orai1, located in the plasma membrane. STIM1 is a sensor that monitors the ER Ca^2+^ stores and increases Ca^2+^ entry into the cell by activating its effector Orai1 when stores are low. This store-operated Ca^2+^ entry (SOCE) system could be a mechanism that matches Ca^2+^ supply to need in the foetoplacental unit. However, there is no available evidence to date that STIM1 and Orai1 are expressed by the villus syncytiotrophoblast (Section 5.3); therefore, this requires confirmation. Because STIM1/Orai1 are widely expressed in organs of mammals postnatally, it would be expected that they are also expressed in the developing foetal organs. The observation that Orai1 was generated in vitro by cells likely to be embryonic stromal cells [203], and by foetal endothelial cells [204], may be preliminary supportive evidence warranting further exploration. Two studies demonstrated that pregnancy-specific beta-1-glycoproteins from the immunoglobulin superfamily, PSG1 [203] and PSG9 [204], increased production of Orai1 and intracellular Ca^2+^ entry (Section 5.3). In response, PSG1 increased trophoblast migration, and PSG9 increased the expression of endothelial nitric oxide synthase (eNOS) and NO production. The suggestion that downregulation of PSG1, PSG9, and Orai1 channel function may be relevant to pre-eclampsia and could have therapeutic potential merits investigation. There is emerging evidence that Orai and STIM may contribute to cardiovascular disease [403], pulmonary hypertension [404], and diabetes [405,406], and the possibilities of pharmacological manipulation are being explored. The findings of these studies may provide insight into placental function.

Because STIM1 and Orai1 are apparently not the SOCE system in the placental syncytium, what is? A strong contender is calmodulin (CaM), which is abundant in terminal microvilli [234]. CaM is not just a Ca^2+^ binder/transporter but also both a Ca^2+^ sensor and effector. CaM loaded with Ca^2+^ controls the two key processes in intracellular Ca^2+^ regulation: Ca^2+^ entry by regulating closure of TRPV6, the principal Ca^2+^ import channel, and Ca^2+^ export by increasing the activity of PMCA1b, the major Ca^2+^-exporting enzyme. CaM is a sensor for increased cytosolic Ca^2+^ because it is activated by Ca^2+^ loading. Activation extends and opens the CaM molecule and separates its N- and C-lobes. These can attach to two sites on the targets TRPV6 or PMCA1b and alter their molecular conformation, in both cases by displacing flexible unstructured peptide sequences in their protein tails (Section 4.2.1 and Section 6.2). It is notable that there are multiple CaM-binding sites both on the polybasic domain of STIM1 downstream of a long unstructured peptide sequence and on the N-terminal of Orai1. It may be that in cells that express STIM1 and Orai1, Ca^2+^-CAM could pull the two proteins together and participate in their interaction. Ca^2+^-CAM may also regulate activation of TRPC channels, which, like TRPV6, have CaM-binding domains in their carboxyl terminals [81].

TRPV6 must be expressed on the villus surface membrane to import Ca^2+^ into the trophoblasts. One mechanism for this is through interaction with cytosolic S100A10/annexinA2 (ANXA2) heterotetramers (Section 4.2.4). The ANXA2 moiety links with phospholipids in the plasma membrane and to the actin cytoskeleton. The S100A10 moiety attaches to a conserved binding site in the TRPV6 C-terminal. Association with the heterotetramer stabilises the channel at the cell surface. It is notable that both *S100A10* and *ANXA2* were among the 115 core genes expressed in term placentas across 14 mammalian species [25], suggesting that the membrane-binding function of the ANXA2/S100A2 heterotetramer has a much wider role than regulating TRPV6 expression. It may be involved in endocytosis or exocytosis and/or the regulation of intracellular transport of other ion channels and transporters.

The review considered evidence for involvement of four peptides and vitamin D in placental Ca^2+^ transport, presented in Section 8, Section 9, Section 10, Section 11 and Section 12. Small studies reported differences in the expression of genes that regulate vitamin D metabolism and Ca^2+^ transport in placentas from pregnancies complicated by pre-eclampsia [341] and gestational diabetes mellitus [341,342] (Section 12.3). Larger studies are required to establish whether vitamin D metabolism is disturbed in GDM and pre-eclampsia and to investigate the functional significance if confirmed. IGF2 and PTHrP are clearly important for normal blastocyst implantation and placental development (paragraph 2 above). Some additional observations are highlighted here.

Unlike PTH, PTHrP is not normally a circulating hormone but has paracrine, autocrine, and intracrine actions. It is abundant in the placenta, expressed very early in embryonic life and needed for blastocyst, placental, and skeletal development but normally not for regulating foetal serum Ca^2+^. PTHrP stimulates placental Ca^2+^ transport but, surprisingly, not through binding to the PTH1R receptor through its amino terminal, which has close homology with the PTH N-terminal, but instead through amino acid sequences in the middle of the molecule with no PTH counterparts. It is speculated that these bind with a receptor that is distinct from the PTH receptor; however, its identity is unknown. Of possible relevance is that there is a lysine-rich sequence in the middle of PTHrP, which may direct PTHrP into the nucleus [270,274]. Perhaps the activation of nuclear transcription factors accounts for many of the activities of PTHrP, which may include activation of placental Ca^2+^ transport.

Calcitonin is the product of the *CALCA* gene. Another protein, CGRP, is also generated from this gene by posttranscriptional modification of the gene transcript. The two proteins have separate receptors, and there is considerable uncertainty about the roles of these proteins in the placenta. The facts and proposals so far are that neither calcitonin nor CGRP is required for foetal–placental Ca^2+^ transfer and that calcitonin produced by uterine endometrial glands from 2 to 6 days of gestation may have a role in blastocyst implantation [1]. An intriguing unanticipated finding was that in mice, Mg^2+^ was significantly reduced in the serum of *calca*-null mothers and foetuses, as well as in the foetal skeleton. There is no obvious link between calcitonin and Mg^2+^ metabolism. One possibility, which has not been considered, is that one of the *CALCA* gene proteins interacts with the TRP carriers TRPM6 and TRPM7 (Section 4.1). Together, these regulate the cellular importation of Mg^2+^ and Mg^2+^-ATP; perhaps the function of calcitonin at implantation is to supply Mg^2+^-ATP. In one published study, the % methylation of the promoter of *CGRP* was higher in cord blood DNA from preterm than term infants (Section 11), but published data on the role of this peptide in the placenta are lacking. Postnatally, CGRP is a vasodilator, and roles in the foetoplacental vasculature were suggested [314] but without any supportive evidence.

Observations on mouse foetuses with CaSR ablation confirm the normal role of the CaSR to suppress PTH secretion by the parathyroid glands. Without CaSR, foetal serum PTH and Ca^2+^ and bone turnover are markedly increased (Section 10). The increased Ca^2+^ must be from bone resorption because placental Ca^2+^ transport was low; no explanation was offered for this observation. It may be that TRPV6 and other Ca^2+^ import channels were inactivated by the raised Ca^2+^ (Section 4.2.1). From case reports, CaSR deficiency was not evident at birth in humans but presented very early in neonatal life with symptomatic hypercalcemia. Already, there were marked parathyroid gland enlargement and bone demineralisation, probably indicating that the problems started in foetal life. The ion and fluid disturbances may be partially corrected in utero via the foetomaternal circulation.

The DOHaD proposal [363] has generated intense research. There is high-quality evidence for associations with increased rates of coronary heart disease, stroke, type 2 diabetes mellitus, adiposity, and metabolic syndrome and early evidence for a link with osteoporosis (Section 14). The sticking block has been to explain the mechanism for the association. As the extent and diversity of epigenetic modifications have become evident, these have emerged as likely mediators. The contribution of transposons to regulation is another aspect to explore [407,408,409]. The genes regulating Ca^2+^ transport are under strong epigenetic control. This underlies attempts to decrease the risk of osteoporosis in later life by manipulating vitamin D and dietary mineral intakes, in addition to lifestyle changes of mothers in pregnancy. There is early evidence that these may succeed, but more large-scale studies that incorporate recent scientific observations are required. The most critical period for adverse modifications of methylation and histones is probably early in embryogenesis (Section 14.2.), suggesting that educational and dietary interventions should be in place antenatally to prepare for pregnancy to reduce risk.

From the above summary, areas of current uncertainty requiring clarification are as follows:The mechanisms of trafficking transporters and ion channels to and from the apical membranes of placental villi.Whether NHERF1 is expressed in placental villi and, if expression is confirmed, the roles of this scaffold in villi.The roles of ezrin in placental villi.Whether calbindin D9k is expressed in the villus syncytium.How Ca^2+^ absorbed for export to the foetus is chaperoned or transported across the trophoblast safely for extrusion.Whether Orai1 and STIM1 are expressed by syncytiotrophoblasts.The roles of Ca^2+^-CaM in mediating SOCE in the syncytiotrophoblasts.The roles of PTHrP mid-molecular residues in regulating DNA transcription, particularly of genes involved in Ca^2+^ transport.Whether CGRP is expressed in the placenta and, if so, its likely roles.Although not relevant to Ca^2+^ transport, it might be of wider interest to find out whether calcitonin or CGRP interacts with TRPM7 or TRPM6.Whether placental metabolism of vitamin D is disturbed in gestational diabetes and pre-eclampsia.Whether pregnancy-specific beta-1-glycoproteins PSG1 and PSG9 increase endothelial NO production and placental blood flow by increasing Orai1 expression.

For the future. The generation of functional multicellular placental organoids from human placental villi or naïve human pluripotential stem cells is a major achievement that will enable studies of normal placental development and villus function and identification of pathologic disturbances, which may lead to pre-eclampsia or IUGR [13,29,32,33,410]. They may also be useful for generating new non-malignant human trophoblast cell lines [411]. Rigorously undertaken analyses of extracellular vesicles derived from the foetal placenta and circulating in maternal blood may uncover predictive markers for placental pathology, enabling early clinical intervention [412,413]. Continued expansion of our knowledge of epigenetic regulation of expression of genes controlling placental Ca^2+^ transfer will guide interventions to protect bone health in later life. If the greatest risk for acquiring epigenetic disorder is in early embryonic life, then dietary and lifestyle improvements should be well established before conception.

## 16. Conclusions

Recent studies have advanced our insight into the mechanisms of placental Ca^2+^ transport considerably, although there are still many gaps to fill. What has emerged is that meeting the high Ca^2+^ demand in the third trimester depends on normal implantation of the blastocyst and placental development in early embryonic life. Ca^2+^ signalling plays an integral part in these processes, and a complex network of Ca^2+^ channels and transporters regulate its availability.

## Figures and Tables

**Figure 1 ijms-26-00383-f001:**
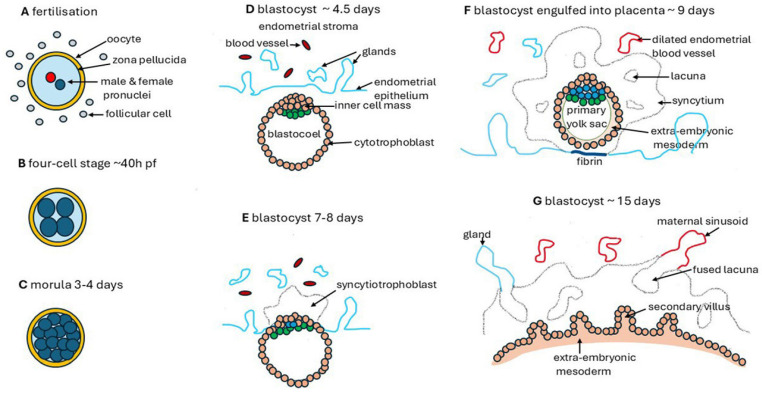
Development and implantation of the blastocyst. Compiled from Burton GJ and Jauniaux E 2021 [6], Mitchell B and Sharma R 2005 [39], Harrison RG 1963 [40], and Sadler TW 2023 [41]; ~, approximately.

**Figure 2 ijms-26-00383-f002:**
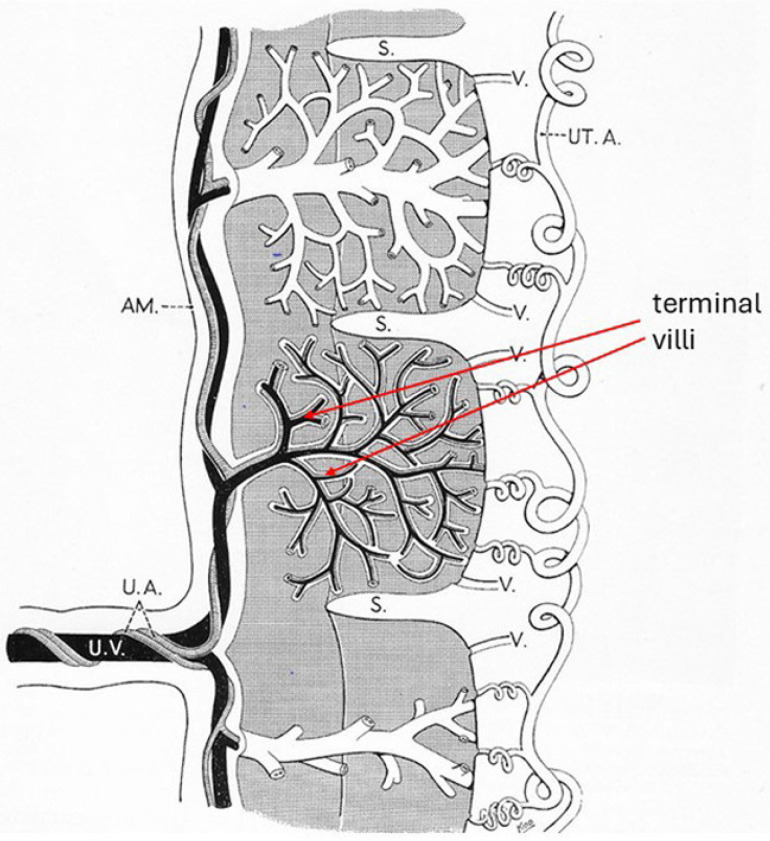
Section of a human placenta showing three cotyledons separated by septa (S). Within each of these is a mass of foetal villi branching from a primary villous stem anchored to the decidua basalis, which forms the maternal placenta. The villi are covered by trophoblasts: an inner layer of cytotrophoblasts and an outer layer of syncytiotrophoblasts. These erode the walls of small uterine spiral arteries, and the blood empties into the intervillous spaces. The highly branched terminal villi (red arrows) float freely in a lake of maternal blood. Nutrients and minerals pass from mother to foetus, but there is no continuity between foetal and maternal circulations. AM, amnion; U.A., umbilical arteries; U.V., umbilical vein; UT.A., uterine artery; V., uterine vein. Source: Harrison RG: A Textbook of Human Embryology 2nd Ed. Blackwell Scientific Publications Ltd. Oxford 1963 [40]. Figure 35 reproduced with permission from Wiley-Blackwell, John Wiley and Sons.

**Figure 3 ijms-26-00383-f003:**
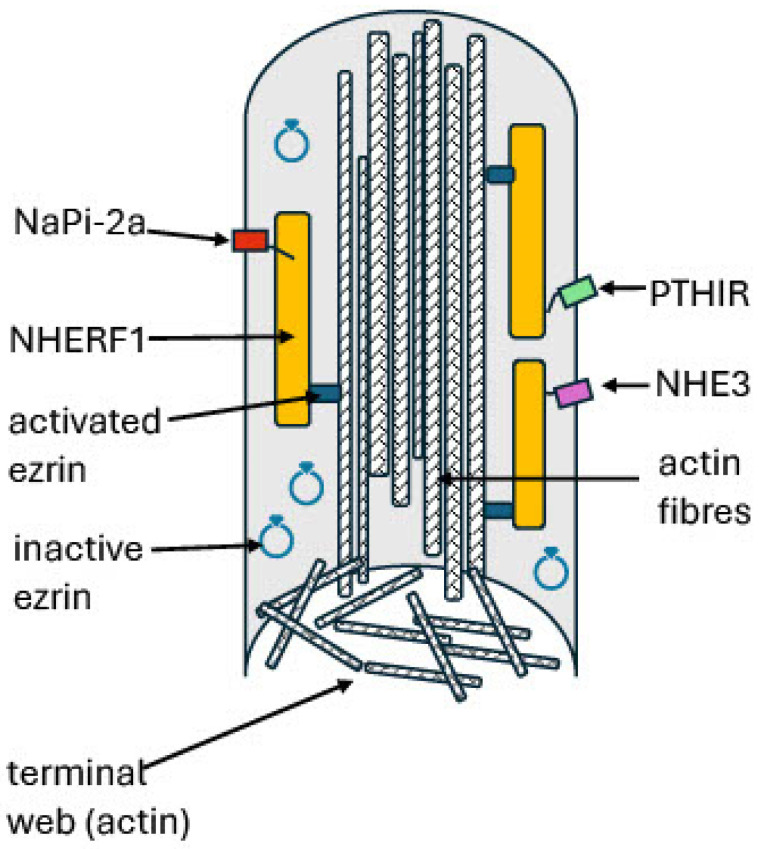
Proposed scheme in which ezrin bridges the scaffolding protein NHERF1 and the actin skeleton, so stabilising proteins in the microvillar membrane and connecting them with cytosolic signalling complexes. (i) In a dormant form, the C-terminal tail of ezrin binds to the N-terminal and closes the molecule. (ii) Phosphorylation of residues located between the N- and C-terminals blocks the association and opens the ezrin molecule. (iii) The freed C-terminal binds to actin; the N-terminal binds to the scaffolding protein NHERF1 associated with the plasma membrane. (iv) NHERF1 connects membrane-associated proteins with transiently assembled cytosolic signalling complexes. NaPi-2a, PTH1R, and NHE3 are shown as examples. NaPi-2a, sodium–phosphate cotransporter 2a; NHE3, sodium–hydrogen exchanger 3; NHERF1, Na^+^/H^+^ exchanger regulatory factor 1; PTH1R, parathyroid hormone 1 receptor.

**Figure 4 ijms-26-00383-f004:**
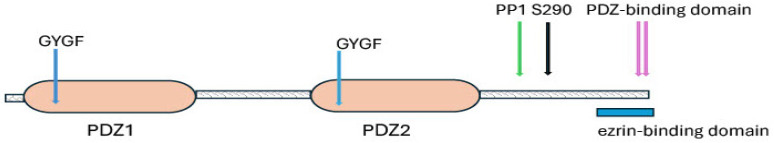
Schematic of Na^+^/H^+^ exchanger regulatory factor 1 (NHERF1) based on the models of Zhang et al. (2019) [62] and Bhattycharia et al. (2019) [63]. GYGF, core PDZ-binding motif; PPI, protein-phosphatase-1-binding site; S290, key phosphorylation site.

**Figure 5 ijms-26-00383-f005:**
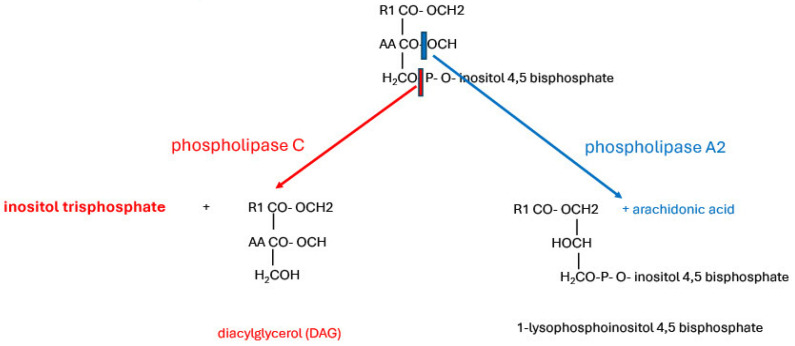
Phospholipase C releases inositol trisphosphate from phosphatidylinositol 4,5 bisphosphate (PIP2). Arachidonic acid is released from PIP2 by phospholipase A2.

**Figure 6 ijms-26-00383-f006:**
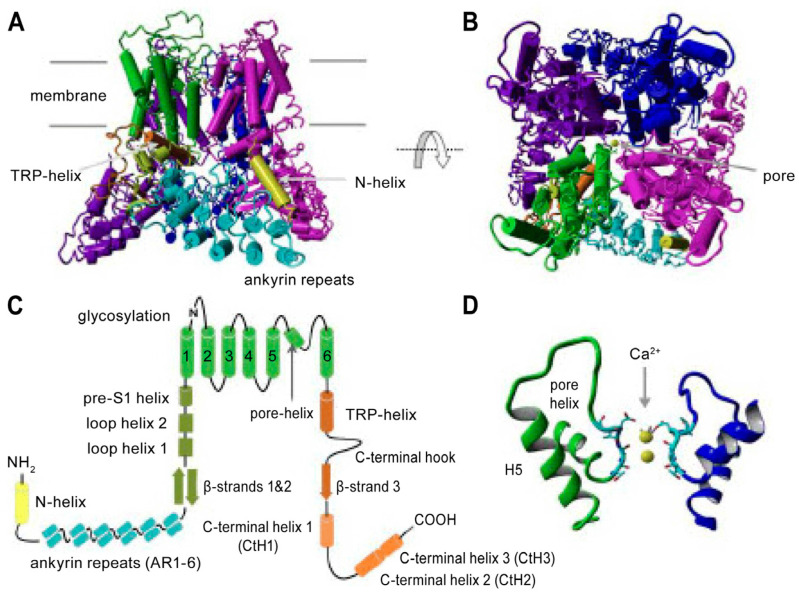
Structure of TRPV6, as deduced from structural studies (PDB codes 5IWK, 5IWP, and 62EF) [132,133,134]. (**A**) Side and (**B**) top views of the four TRPV6 monomers, displayed as ribbons with α-helices and β-sheets as cylinders and arrows, respectively. Three monomers are coloured uniformly (purple, blue, and dark pink); one monomer is differentially coloured to highlight crucial regions: N-terminal helix (yellow), ankyrin repeats (cyan), intracellular N-terminal region (dark green), membrane region (green), and intra-cellular C-terminal region (orange). The Ca^2+^ ion in the pore is displayed as a yellow ball. (**C**) Schematic overview of secondary structural elements and other structural features. Colour coding as in panels (**A**,**B**). Note that the C-terminal helices 1–3 (CtH1–3; light orange) are not observed in the 5IWK structure displayed in panels (**A**,**B**,**D**). (**D**) Close-up of the TRPV6 pore-forming region. The ion selectivity filter forming the side chains of D542 and surrounding residues are shown in ball-and-stick representation, colour-coded on a by-atom-type basis. The carboxylic groups lining the pore are clearly visible. The ion selectivity filter is the narrowest part of the channel. This figure was published in *Advances in Clinical Chemistry*, vol. 113, Walker V, Vuister GW, Biochemistry and pathophysiology of the transient potential receptor vanilloid 6 (TRPV6) calcium channel, pp 43–100, 2023 [115]. Copyright Elsevier. Reproduced with permission.

**Figure 7 ijms-26-00383-f007:**
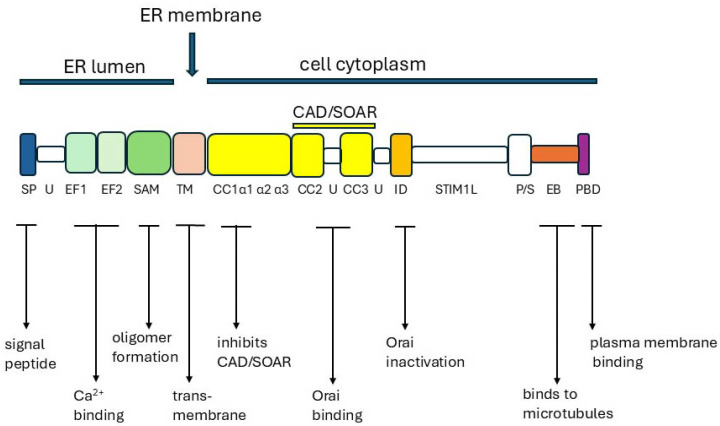
The domains of STIM1 and their functions [61,185,187]: SP, signal peptide; EF1 and EF2, EF-hand domains; SAM, sterile alpha motif; TM, transmembrane domain; CC1, CC2, and CC3, coiled-coil domains; CAD, CRAC activation domain; SOAR, STIM-Orai activating domain; ID, inactivation domain; STIM1L, a peptide insert; P/S, proline/serine-rich peptide; EB, EB1-binding domain; PBD, polybasic domain; U, peptide sequences.

**Figure 8 ijms-26-00383-f008:**
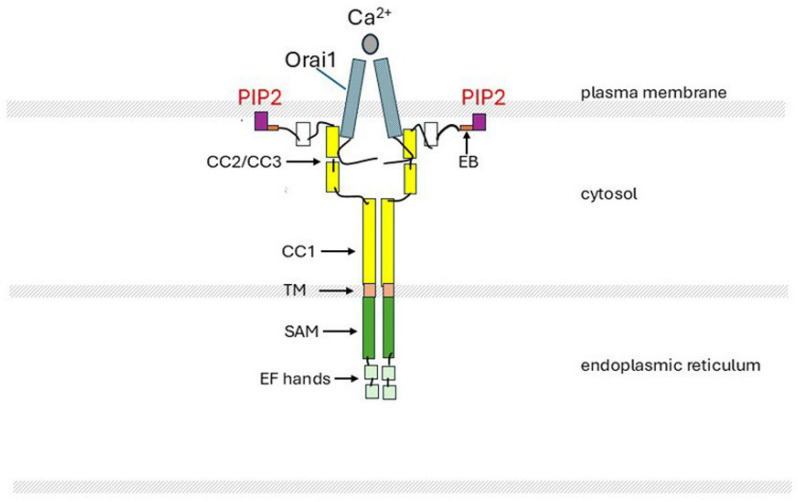
Binding of activated STIM1 to Orai1 to open the Orai1 channel for Ca^2+^ influx. SAM, sterile alpha motif; TM, transmembrane domain; CC1, CC2, and CC3, coiled-coil domains; PBD, polybasic domain; PIP2, phosphatidylinositol 4,5 bisphosphate.

**Figure 9 ijms-26-00383-f009:**
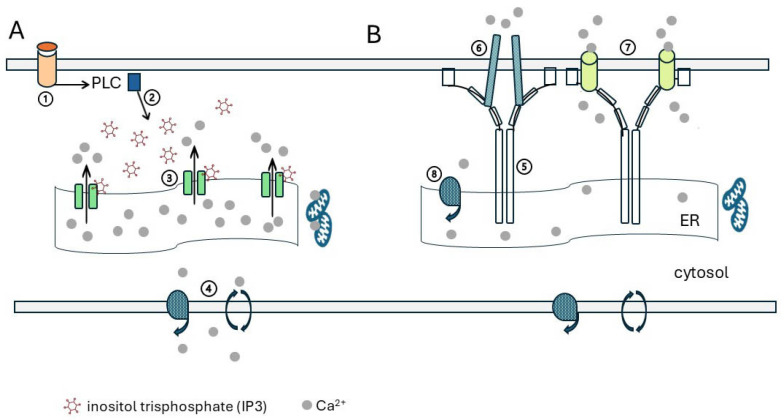
Store-operated calcium entry (SOCE). (**A**) (1) Agonist stimulation of a membrane receptor (2) activates phospholipase C (PLC), releasing inositol trisphosphate (IP3), and (3) IP3 activates the IP3 receptor in the ER membrane, leading to release of Ca^2+^ into the cytosol to activate signalling cascades. Some Ca^2+^ is taken into mitochondria, and (4) surplus Ca^2+^ is extruded from the cytosol by plasma membrane Ca^2+^-ATPase 1 (PMCA1) and/or the NCX Na^+^/Ca^2+^ exchanger. (**B**) (5) Ca^2+^ depletion in the endoplasmic reticulum (ER) activates the Ca^2+^ sensor STIM1, and (6) activated STIM1 binds to Orai1 in the plasma membrane, opening the channel for Ca^2+^ entry. (7) Activated STIM1 can also associate with transient receptor C (TRPC) proteins, opening their channels for Ca^2+^ entry. (8) Imported Ca^2+^ is transported into the ER by sarcoendoplasmic reticulum Ca^2+^ ATPase (SERCA) to replenish the ER stores.

**Table 1 ijms-26-00383-t001:** Events in the formation of the mature human placenta [6,39,40,41].

^ꝉ^ Time pf.	^ꝉꝉ^ Figure Number	Events	Source of Ca^2+^
0	1A	Fertilisation. Sperm releases phospholipase zeta (PLCζ), leading to Ca^2+^ release and cell division.	Endoplasmic reticulum (ER) stores in oocyte
Approx. 36 h		Zygote cleaved to two cells.	^ꝉꝉꝉ^? ER stores
Approx. 40 h	1B	Four-cell stage.	^ꝉꝉꝉ^? ER stores
3–4 d	1C	Morula, a sphere of 12–16 cells enclosed within the ovarian zona pellucida, reaches junction of fallopian tube with uterus.	^ꝉꝉꝉ^? ER stores
Approx. 4.5 d	1D	Cavity in morula (blastocoel) fills with fluid, forming a blastocyst. The outer cells form a spherical wall of trophoblasts and inner cells accumulate at one pole as an inner cell mass, which forms the embryo.	Uterine secretions diffusing into blastocoel
Approx. 5 d		Start of blastocyst implantation in uterine endometrium induces decidualisation of endometrial glands.	Uterine secretions via blastocoel
Approx. 7–8 d	1E	Blastocyst partially implanted. Inner cell mass now has two layers: epiblast and hypoblast. The amniotic cavity develops in the epiblast, and hypoblast cells extend around the blastocoel, forming the primary yolk sac. The trophoblast wall produces the following: i) a cellular syncytium, which surrounds the blastocyst, invades the endometrium, and contains lacunae and ii) a lining of extra-embryonic mesoderm, which later fuses with the syncytium to form the foetal chorion.	Decidual fluid-? via yolk sac
Approx. 9 d	1F	Blastocyst is fully implanted within the endometrium, syncytial lacunae extend into the endometrial stroma, and columns of syncytial villi start to grow towards the basal layer of uterine epithelium (anchoring/stem villi).	Decidual fluid-? via yolk sac
Approx. 12 d		Syncytial clefts fuse with endometrial glands containing decidual secretion.	Decidual fluid (uterine secretion) histotrophic nutrition
By 14 d		Syncytial trophoblasts breach uterine vessels, and blood escapes into the syncytial lacunae. Syncytial columns now anchor the foetal chorion to the basal layer of the uterine epithelium (anchoring/stem villi). Finger-like processes of cytotrophoblasts grow into the syncytium, forming primary villi that surround the blastocyst. Villi then degenerate, except those adjacent to the uterus, which form the chorion frondosum and ultimately the discoid placenta.	Decidual fluid, histotrophic nutrition, and some maternal blood
15 d	1G	Extra-embryonic mesoderm invades the primary villi, converting them to secondary villi, now classed as chorionic villi, and forms a connecting stalk to the embryo, which will become the umbilical cord. The chorionic villi are suspended in spaces of maternal blood.	Decidual fluid, histotrophic nutrition, and some maternal blood
15–20 d		Foetal capillaries develop in the secondary villi, which then become tertiary villi.	Histotrophic nutrition and some maternal blood
10–12 weeks		Maternal circulation to placenta established.	Maternal blood and some histotrophic nutrition
By 20 weeks	Figure 2	Mature placenta formed with free-floating terminal villi—the dominant absorption structures.	Maternal blood
From 20 weeks		Exponential growth of placenta to term.	Maternal blood

^ꝉ^ pf., postfertilisation. ^ꝉꝉ^ Figure numbers 1A to IG are the numbers for these stages as depicted in Figure 1. ^ꝉꝉꝉ^? Indicates that this is the likely source but conjectural. Approx., approximately.

**Table 2 ijms-26-00383-t002:** Human mutations relevant to placental calcium transport.

Disorder [References]	Gene	Protein	Plasma Ca^2+^	PTH	Bone Deformity	Birth Weight
Isolated PTH deficiency [343,344]	*PTRH*	PTH	↓	↓	No	Normal?
Jansen’s metaphyseal chondrodysplasia [345,346,347,348]	*PTH1R*	PTH/PTHrP receptor	↑	↓	Yes	↓
Pseudohypoparathyroidism 1A [345,349,350,351]	*GNAS* from mother	Gsα	↓	↓	Yes	Normal?
Pseudo-pseudohypoparathyroidism [345,349,350,351]	*GNAS* from father	Gsα	Normal	Normal	Yes	Normal?
I-cell disease [352]	*GNPTAB*	N-acetylglucosamine phosphotransferase	↓ or normal	↑	Yes	Low
Severe neonatal hyperparathyroidism [294,301,352]	*CaSR* inactivating	Calcium sensor receptor	↑↑	Not suppressed	Yes	Usually diagnosed postnatally
Silver–Russell syndrome [267,353]	*IGF2*	IGF2 deficiency	Normal	Normal	Yes	Low
Temple’s syndrome [250,354]	*IGF2*	IGF2 deficiency	^ꝉ^ Normal?	Normal?	No	Low
Beckwith–Wiedemann syndrome [267,355]	*IGF2* loss of imprinting	IGF2 excess	Normal?	Normal?	Hemihypertrophy	Increased
Transient neonatal hyperparathyroidism [155,156,157,158,159]	*TRPV6*	Transient potential receptor vanilloid 6 (TRPV6)	↓ or normal	↑	Yes	Normal or low

^ꝉ^ Normal? assumed normal because there is no reported information. Add to legend: ↑↑ large increase, ↑ increase, ↓ decrease.

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
