# Peer review of "The Molecular Biology of Placental Transport of Calcium to the Human Foetus"

_ijms, 2025, doi:10.3390/ijms26010383_

Round 1
Reviewer 1 Report
Comments and Suggestions for Authors
In the present work, Walker tries to review of placental calcium transport in four sections. Section A describes placental development, and section B covers calcium carriers and transporters which control calcium. In addition, section C covers the effects of peptides and hormones on placental calcium transport, and section D is a brief overview of the proposed developmental origins of health and disease in relation to postnatal bone development. However, there are some questions that should be explained.
Major concerns
1. As a Review article, recent related articles should be referred. However, lots of recent related articles have not been referred. For example,
Varshney S, Adela R, Kachhawa G, Dada R, Kulshreshtha V, Kumari R, Agarwal R, Khadgawat R. Disrupted placental vitamin D metabolism and calcium signaling in gestational diabetes and pre-eclampsia patients. Endocrine. 2023;80(1):191-200.
Mariacarmela G, Milena E, Sveinbjorn G, Daniel H, Maurizio M. Placental protein 13 dilation of pregnant rat uterine vein is endothelium dependent and involves nitric oxide/calcium activated potassium channels signals. Placenta. 2022;126:233-238.
Xing P, Hong L, Yan G, Tan B, Qiao J, Wang S, Li Z, Yang J, Zheng E, Cai G, Wu Z, Gu T. Neuronatin gene expression levels affect foetal growth and development by regulating glucose transport in porcine placenta. Gene. 2022;809:146051.
Rengarajan A, Austin JL, Stanic AK, Patankar MS, Boeldt DS. Mononuclear Cells Negatively Regulate Endothelial Ca2+ Signaling. Reprod Sci. 2023;30(7):2292-2301.
2. Figure 1 should be revised. There are some wrongs. The male pronucleus is not right, which should not have a tail. In the implantation site, there should have syncytial plaque. In addition, uterine epithelium should be revised.
3. Figure 2 should be revised. The mature placenta should include fetal arterial and venous blood vessels, and uterine arterial and venous blood vessels, but there are only uterine arteries.
4. Figure 3 should be explained more detail.
5. Figure 6 should use a high quality figure, and there are some blurred.
6. English grammar and writing style should be checked and revised throughout the manuscript.
Minor concerns
1. The whole writing style of this manuscript is not suitable for IJMS, including the format of all references. ‘2008’ is what in the head title?
2. The writing style of author’s name is not right.
3. Abstract section should be revised, and a summary should be in the end of the Abstract.
4. Keywords should be revised, and some keywords are not suitable.
5. ‘Ca2+’, ‘2+’ should be in superscript, there are so many that are not in superscript. Please check these throughout the manuscript, including Figure 8.
6. Page 3, ‘(Jackson 1992).’.
7. Page 7, ‘(Figure 4.)’, ‘Ca2+trigger’, ‘Ca2+and’.
8. Page 8, ‘Ca2= entry’, ‘(Figure 5.)’, ‘Ca2+oscillations’.
9. Page 13, ‘[165,169, 170,164, 163].’.
10. Page 14, ‘[Reviewed 187-190].’.
11. Page 15, ‘Ca2+atoms’.
12. Page 16, ‘(Fig.8)’, ‘Figure 8 Figure 8.’.
13. Page 18, ‘[Ca2+]ER by SERCA’, ‘Ca2+is’.
14. Page 22, ‘Ca2+transport [291,292,293],’.
15. Page 23, ‘Ca 2+ (Section 4).’, ‘Ca 2+and’.
16. Page 26, ‘Ca2+status’.
17. Page 28, ‘[2,17, 372, 373].’.
18. Page 29, changes ‘Summary and Discussion’ to ‘Discussion’.
19. Page 32, the numbers 1., 2… are not right.
Comments on the Quality of English LanguageThe English could be improved to more clearly express the research.
Reviewer 2 Report
Comments and Suggestions for Authors
This review summarizes the molecular biology of the placental transport of calcium to the human fetus and gives new insight into future directions that need to be analyzed. However, a few revisions are necessary to clarify the research findings. The author must address a few points.
1) Page 1: By ~ 35 weeks of gestation, around 300mg of Ca2+ are transferred daily [1], and third that it matches the supply to the changing and increasing needs of the fetus [2, 3, 4]. Before~16 weeks of gestation, Ca2+ is provided by uterine fluid which contains secretions from decidual glands in the uterine endometrium [5,6].
Please change ‘~’ with ‘approximately’ or similar. Review the entire text.
2) The ‘Introduction’ section is concise and reflects the aim of the study.
3) Page 3: For detailed information, refer to [5,6, 33, 35]
Please remove this proposition or describe the mentioned references in another paragraph.
4) For Figure 1 and Table 1 mention the cited sources
5) For Table 1, please create a note where you mention the origin of the abbreviation ‘ER’
6) Reorganize Table 1 in a more compact manner for a better understanding
7) Spiral branches of the maternal uterine arteries empty into the intervillous spaces. Blood percolates around the villi and the leaves via uterine veins. Two fetal arteries carry blood from the villi to the fetus via the umbilical cord, where they lie alongside a single vein carrying blood from the fetus to the maternal venous system. There is never any continuity between maternal and fetal circulations in an undamaged placenta.
Please introduce the sources of this paragraph.
8) Topic 3 ‘Ca 2+ release from reticular endothelial stores’ and topic 4 ‘Ca2+ importation across the plasma membrane’ should be re-arranged. Some subchapters are too extensive and should refer only to their activity in the process. Consider excluding the description of biochemical structure and molecular mass.
9) Clarsen, Roberts, Hamark et al. [176] demonstrated for the first time that SOCE occurred in term human placenta, but not first trimester placenta, and that the channel was inhibited by GdCl3, NiCl2, CoCl2 or MSKF96365, a channel-blocking chemical, but not by nifedipine. Investigations of a family with inherited severe combined immunodeficiency (SCID) identified the CRAC channel as Orai1 [177-179]
It would be helpful to provide more information about the results of these mentioned references.
10) The STIM-Orai activating region (named SOAR) is located at two of the three highly conserved coiled-coil (CC) domains CC2/CC3 on the cytosolic side of the ER membrane. Positively charged lysine residues in the polybasic domain (PBD) at the C-terminal interact electrostatically with acidic phospholipids and acyl chains in the plasma membrane, including PIP2, and localize the activated STIM proteins at the ER-plasma membrane (PM).
Please introduce the sources of this paragraph.
11) Stromal interaction molecule 1 and 2 (STIM) have become in the past few years a marker in detecting hypertension, obesity or diabetes. Mention through other studies their implications.
12) There is only one published report [199], summarised below, demonstrating the expression and operation of the Orai/STIM partnership in the placenta… The selective Orai1 inhibitor (MRS1845) weakened the migration-promoting effect mediated by PSG1 by suppressing Akt signaling [199].
In the mentioned study, they show extensive results of their finding. I suggest mentioning their results and their association with preeclampsia would be interesting.
13) Regarding point number 12, there is also another interesting study that explores another pregnancy-specific glycoprotein, PSG9 changes in patients with preeclampsia and the effects and underlying mechanisms on calcium homeostasis that should be mentioned.
Qin, Y.; Meng, Q.; Wang, Q.; Wu, M.; Fang, Y.; Tu, C.; Hu, X.; Shen, B.; Chen, H.; Xu, X. Pregnancy-Specific Glycoprotein 9 Enhances Store-Operated Calcium Entry and Nitric Oxide Release in Human Umbilical Vein Endothelial Cells. Diagnostics2023, 13, 1134.
https://doi.org/10.3390/diagnostics13061134
14) I recommend reorganizing section C: ‘Effects of Peptides and Hormones on Placental Ca2+ transport. This section should highlight more information about the importance of each mentioned hormone on placental and fetal growth, without using such an extensive biochemical information.
15) ‘PTHrP is increased in cord blood in fetal growth restriction.’
Please explain the mechanism behind it.
16) Table 2 is well-structured and concise
17) ‘The COPSAC2010 trial observed similar increases in WBBMC and aBMD at age 6y. These findings suggest that higher doses of antenatal vitamin D supplementation have beneficial effects on offspring skeletal mineralization.’
A statistic of efficacy/ effectiveness would be helpful.
18) Future directions are concise and highlight the takeaways of each section
19) Regarding the bibliography, try to change older cited articles before 2008 with newer evidence